# Clump sequencing exposes the spatial expression programs of intestinal secretory cells

Rita Manco[1,4], Inna Averbukh[1,4], Ziv Porat [2], Keren Bahar Halpern[1], Ido Amit [3] & Shalev Itzkovitz [1✉]

Single-cell RNA sequencing combined with spatial information on landmark genes enables reconstruction of spatially-resolved tissue cell atlases. However, such approaches are challenging for rare cell types, since their mRNA contents are diluted in the spatial transcriptomics bulk measurements used for landmark gene detection. In the small intestine, enterocytes, the most common cell type, exhibit zonated expression programs along the crypt-villus axis, but zonation patterns of rare cell types such as goblet and tuft cells remain uncharacterized. Here, we present ClumpSeq, an approach for sequencing small clumps of attached cells. By inferring the crypt-villus location of each clump from enterocyte landmark genes, we establish spatial atlases for all epithelial cell types in the small intestine. We identify elevated expression of immune-modulatory genes in villus tip goblet and tuft cells and heterogeneous migration patterns of enteroendocrine cells. ClumpSeq can be applied for reconstructing spatial atlases of rare cell types in other tissues and tumors.

[1] Department of Molecular Cell Biology, Weizmann Institute of Science, Rehovot, Israel. [2] The Flow Cytometry Unit, Life Sciences Faculty, Weizmann Institute of Science, Rehovot, Israel. [3] Department of Immunology, Weizmann Institute of Science, Rehovot, Israel. [4] These authors contributed equally: Rita Manco, Inna Averbukh. ✉email: shalev.itzkovitz@weizmann.ac.il

Many tissues such as the liver, the intestine, and the kidney, are composed of structured anatomical units[1]. Spatially varying concentrations of oxygen, nutrients, and morphogens along these units dictate distinct gene expression signatures for cells residing at different spatial coordinates, a phenomenon termed "zonation"[1]. Approaches to spatially reconstruct zonation patterns combine scRNAseq with spatial expression profiles of landmark genes characterized by RNA in situ hybridization[2–6]. An alternative approach, when no prior knowledge of landmark gene candidates exists, entails the spatial measurements of the complete transcriptome of small tissue regions, isolated using laser capture microdissection (LCMseq)[7]. While these approaches successfully reconstruct the zonation patterns of the major cell types in a tissue, rare cell types are more challenging, since their transcript contents are diluted in the spatial measurement techniques.

A recent approach for sequencing pairs of attached cells enabled reconstructing the zonation patterns of liver endothelial cells, by utilizing the landmark genes of their attached hepatocytes[8]. This approach relies on the prospective isolation of mixed pairs, requiring unique surface markers for the cell types of interest, markers which do not generally exist. Reconstructing zonation patterns of rare tissue cells, therefore, remains an open challenge.

In the small intestine, epithelial cells operate in repeating crypt-villus units (Fig. 1a). Crypt-harboring stem cells and progenitors, supported by a Paneth cell niche, continuously divide to generate differentiated cells[9]. Around 90% of the differentiated epithelial cells are absorptive enterocytes. LCMseq-guided single-cell reconstruction revealed profound zonation of enterocyte gene expression along the villus[7]. Additional secretory epithelial cell types include mucus-producing goblet cells[10,11] (~8%), hormone producing enteroendocrine cells[12] (~1%) and chemosensory tuft cells[13,14] (~1%). These rare cell populations are important for the protection of the tissue and for communication with other stromal cell types and with other organs[15]. Using a reporter mouse model, Clevers and colleagues reconstructed the temporal expression programs of enteroendocrine cells along the crypt-villus axis[12], however, zonated expression patterns of goblet and tuft cells are unknown.

To overcome the limitations in reconstructing spatial expression profiles of rare cells, we present ClumpSeq, an approach for sequencing small clumps of attached tissue cells. Sequencing clumps increases the capture rate of rare cell types without the need for antibody enrichment, and utilizes the spatial information of the major tissue cell type. We use this approach to reconstruct spatial maps of all intestinal secretory epithelial cell types along the crypt-villus axis, revealing zonated immune-modulatory programs and heterogenous migration patterns.

## Results

### ClumpSeq enables reconstructing spatial expression patterns of rare cell types

ClumpSeq entails the sub-optimal dissociation of the epithelium into small clumps of 2–10 cells (Fig. 1a, b, Supplementary Fig. 1a, b) and sequencing of these clumps using scRNAseq protocols. Analysis of clumps increases the capture rate of rare epithelial cells, while avoiding the need for dedicated surface markers and for massive numbers of sequenced cells. The enterocyte transcriptome in each clump enables inferring the clump location along the crypt-villus axis (Fig. 1c). Such information facilitates extracting large panels of spatially-varying landmark genes that are specific to the rare secretory cell type of interest, enabling spatial reconstruction of their entire transcriptome by integrating single-cell data (Fig. 1d).

We applied sub-optimal tissue dissociation using EDTA without the addition of commonly used dissociation enzymes ("Methods") and stained for DNA content using Hoechst dye. We used fluorescence-activated cell sorting (FACS) to select clumps based on the DNA content (Fig. 1b, Supplementary Fig.1b). We sorted the clumps into 384-well plates and applied the MARS-seq protocol[16] for sequencing their transcriptomes. The resulting clumps exhibited zonation patterns, as evident by the distinct expression in different clumps of the crypt (Fig. 2a), bottom villus (Fig. 2b), mid-villus (Fig. 2c) and villus tip enterocyte genes (Fig. 2d). In addition to these zonated enterocyte genes, many clumps exhibited mRNAs of secretory genes, attesting to the successful capture of goblet cells (Fig. 2e), enteroendocrine cells (Fig. 2f), tuft cells (Fig. 2g) and Paneth cells (Fig. 2h). The fraction of cells with secretory transcripts was significantly higher in the larger clumps compared to the 2-cell clumps (Fig. 2i–l, Mann–Whitney $p < 10^{-6}$, Supplementary Fig. 1c). The expression of villus tip markers and crypt markers were strongly anti-correlated (Fig. 2m, $R_{Spearman} = -0.82$, $p < 10^{-6}$), and Paneth cells markers were almost exclusively found in crypt clumps (94.6%, Fig. 2n). To further verify that single cells do not form clumps in solution, we demonstrated that the percentages of clumps immediately after tissue dissociation and after 3 h of incubation had little variation (clumps dissociation protocol, 7.79% of clumps at time 0 h vs 7.95% of clumps after 3 h, $p = 0.12$, Supplementary Fig. 2). Thus, the sequenced clumps consist of cells that were attached in the tissue, rather than dissociated cells that have come into contact after tissue dissociation.

We developed a geometric algorithm for classifying the clumps into their cell-type constituents (Fig. 2o, Supplementary Fig. 3a, "Methods"). We reconstructed the location of each clump using enterocyte landmark genes (Supplementary Fig. 3b–d, Supplementary Data 1, Supplementary Data 2)[7]. We next used scRNAseq datasets[12,17,18] to identify genes that are expressed at high levels in the secretory cells of interest and at low levels in enterocytes. The transcripts of these genes in clumps likely originate from secretory cells rather than from enterocytes. Among these secretory cell-specific genes, we identified sets of landmark genes that decreased or increased in expression from the crypt bottom clumps toward the villus tip clumps ("Methods", Supplementary Data 1). These constitute secretory landmark genes that can be used for single-cell reconstruction ("Methods", Supplementary Figs. 4–6).

**Tip goblet cells exhibit enriched immune modulatory programs**. We first applied our approach to goblet cells, the most abundant secretory cell type in the intestine[10]. Our ClumpSeq data included 1,140 high-confidence goblet cell -containing clumps. We extracted 371 landmark genes (Supplementary Data 1) and used them to infer the position of single goblet cells, sequenced with the MARS-seq protocol (Supplementary Fig. 7). We grouped these goblet cells into 5 zones, ranging from crypt goblet cells to villus-tip goblet cells and computed a zonation table by averaging the cells within each zone (Supplementary Data 3, "Methods"). We found that around 30% of the highly expressed genes in goblet cells were significantly zonated (1187 out of 3967 genes expressed to levels above $5 \times 10^{-5}$ had zonation $q$-value < 0.25, Fig. 3a). We used single-molecule fluorescence in-situ hybridization (smFISH, Fig. 3b, e, Supplementary Fig. 8) and the ClumpSeq data (Supplementary Fig. 9a) to validate the accuracy of our goblet zonation reconstruction (Spearman correlation between the ClumpSeq and single-cell reconstructed zonation profiles of a validation set, $R = 0.6$, $p = 8 \times 10^{-4}$, "Methods").

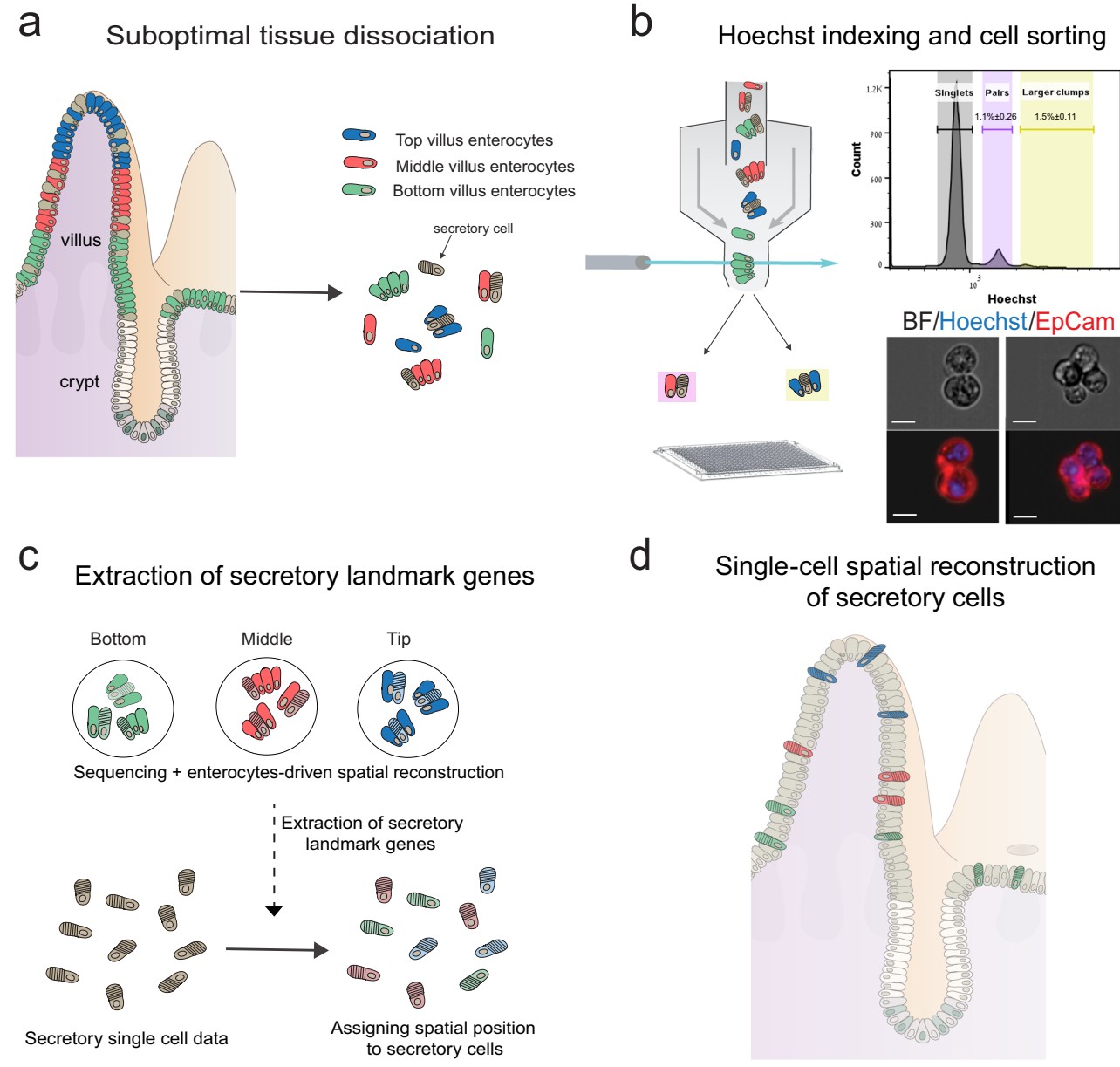

**Fig. 1 Schematic representation of the experimental design. a** The intestinal tissue is suboptimally dissociated to generate clumps. **b** Clumps are enriched with FACS, based on Hoechst DNA staining; the histogram shows ImageStream quantification of the clumps' nuclear DNA content ($n = 3$ mice). Source data are provided as a Source data file; bottom shows an example of a pair (left) and a 4-cell clump (right). Scale bar, 10 μm. **c** The position of clumps is computationally inferred by the enterocyte transcriptome, and spatial landmark genes for specific secretory cells are retrieved. **d** These are used to infer the location of single sequenced secretory cells, enabling zonation reconstruction.

Most genes encoding mucus components exhibited increased expression toward the villus tip (Fig. 3c). An exception was *Agr2*, a gene hypothesized to be secreted in molar quantities with *Muc2*[10], which we found to be inversely zonated toward the crypt. This finding of an anticorrelated zonation profile supports the hypothesis that *Agr2* may play additional roles in goblet cells, for example in goblet cell maturation at the crypt[19]. Our analysis revealed zonation of ligands, receptors[20], and transcription factors[21] in goblet cells (Supplementary Fig. 9b–d), including tip-enriched expression of the immediate-early genes *Jun* and *Atf3*. Gene sets related to RNA polymerase, splicing and ribosome were zonated toward the crypt and villus bottom (Fig. 3d, Supplementary Data 4) largely overlapping the functional zonation previously measured for crypt and bottom villus enterocytes[7]. Goblet genes at the villus tip were enriched in

cytoskeleton and tight junction genes, resembling the structural changes previously observed for tip enterocytes[7] (Supplementary Data 4). Notably, villus-tip goblet cells exhibited enriched immune-modulatory programs, including interferon-alpha and interferon-gamma responses (Fig. 3d, Supplementary Data 4). Interferon-alpha (IFNα) and interferon-gamma (IFNγ) responses were also enriched in tip enterocytes, however, the identity of the tip-enterocyte and tip-goblet cell genes was largely distinct (Supplementary Fig. 10). Tip goblet cell IFNγ genes included the immune checkpoint target gene *Ido1* (Fig. 3a, e), previously shown to have immunosuppressive effects[22]. Tip enterocyte IFNγ genes included the viral response receptor *Ddx58*[23]. The different clusters suggest that all cells at the tip react to IFNγ, but through distinct mechanisms that are cell-type specific. Consistent with these tip programs, the most goblet cell tip-zonated mucus genes

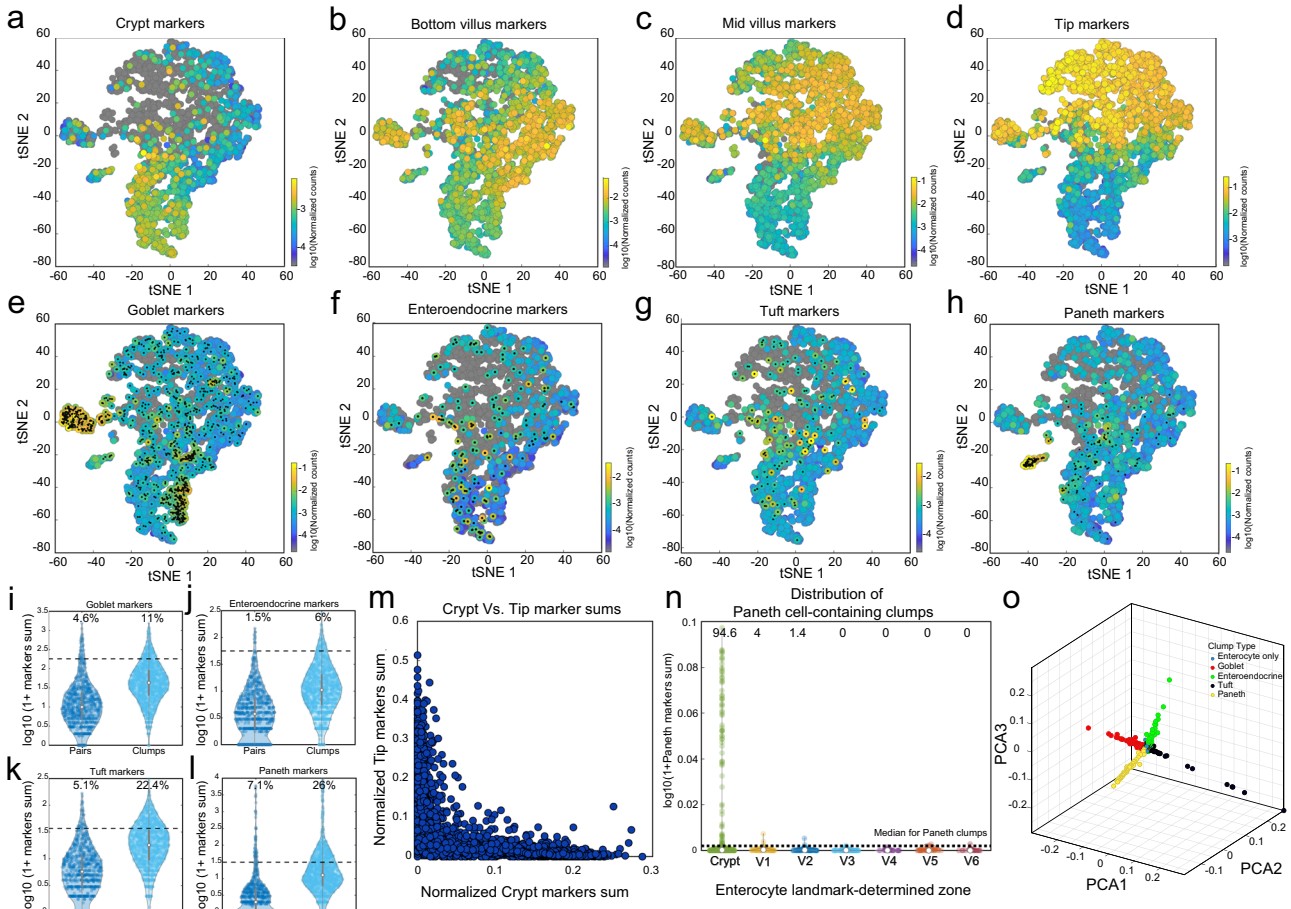

**Fig. 2 ClumpSeq yields tissue fragments consisting of secretory and non-secretory cells. a–d** tSNE plots of sequenced clumps, colored by log10 of summed expression of zonated enterocyte markers: **a** the crypt genes Mki67, Ccnb1, Ccnd1, Mcm2, Pcna, and Olfm4; **b** the bottom villus genes Nlrp6, Lypd8, Il18, Reg1 and Reg3a; **c** the mid-villus genes Slc5a1, Slc2a5, Slc2a2, Slc7a7, Slc7a8, Slc7a9; **d** the villus tip genes Ada,Nt5e and Slc28a2, Creb3l3, Apoa1, and Apob. **e–h** tSNE plots highlighting clumps containing secretory cells, marked by black dots. Plots colored by log10 of summed cell type marker genes (Supplementary Data 7, "Methods") for **e** Goblet, **f** Enteroendocrine, **g** Tuft and **h** Paneth cells. **i–l** Large clumps increase the capture rate of secretory cells. Shown are violin plots of summed secretory derived transcripts (expressed in over 50% of single secretory cells with a mean over 5 fold higher than in enterocytes) in pairs compared to larger clumps (Supplementary Data 6) for **i** Goblet, **j** Enteroendocrine, **k** Tuft and **l** Paneth specific genes. Only crypt pairs and clumps were used to minimize effects from zonated genes. White circles are medians, gray boxes mark the 25–75 percentiles. Dashed horizontal lines indicate the median value in the respective single secretory cell type (Supplementary Fig. 7). Numbers show the percent of clumps above this threshold, which most probably contain the respective secretory cell type. n pairs = 2926, n clumps = 1862, examined over five independent experiments. **m** Crypt and villus-tip enterocyte marker genes are not found in the same clumps, indicating the clumps did not form after tissue dissociation. **n** Violin plot of log10 of 1+summed paneth markers (Supplementary Data 7) in all large (more than two cells) clumps, stratified by inferred zone. White circles are medians. Dashed horizontal line indicates the median value in paneth containing large clumps. **o** Geometric classification of clumps. Representation of clumps in PCA space based on the type-specific markers summed expression. Enterocyte-only clumps are at the origin, each ray contains a different secretory cell type ("Methods", Supplementary Fig. 3e, f, Supplementary Data 7).

were *Muc3* and *Muc4* (Fig. 3c), transmembrane mucins that act as bacterial receptors[24]. Our zonation reconstruction thus points to immune-specialization of goblet cells at the tips of the villi.

**ClumpSeq reveals zonation of Tuft cell programs**. Tuft cells are rare chemosensory epithelial cells with important functions in mediating type-2 immunity, most notably against intestinal worm infections[13]. Recent work demonstrated diversity of individual tuft cells[17], but spatial heterogeneity of tuft cells along the crypt-villus axis has not been explored. Our ClumpSeq data included 146 tuft cell-containing clumps, from which we extracted 352 landmark genes (Supplementary Data 1). We used these landmark genes to reconstruct a dataset of 144 single sequenced tuft cells assembled from previous work[17] and 159 tuft cells sequenced with MARS-seq (Supplementary Data 3, "Methods"). We found that around

17% of the highly expressed genes in tuft cells were significantly zonated (1240 out of 7360 genes expressed at levels above $5 \times 10^{-5}$ had *q*-value < 0.25, Fig. 4a). We used smFISH (Fig. 4b–d, Supplementary Figs. 11 and 13b–d) and ClumpSeq data to validate the accuracy of our tuft zonation reconstruction (Spearman correlation between the ClumpSeq and single-cell reconstructed zonation profiles of a validation set, $R = 0.62$, $p = 5 \times 10^{-4}$, "Methods").

Tuft cells at the crypt expressed the transcription factor *Sox4*, previously shown to be important for Tuft cell specification[25]. Tuft cells at the villus tip expressed the fatty acid binding protein 1 (*Fabp1*) and the succinate receptor 1 (*Sucnr1*), suggested to act as a sensor for infectious agents[26] (Fig. 4a–d, Supplementary Fig. 12). Tip tuft cells also expressed *Il17rb*, the receptor for *Il25*, a tuft-specific cytokine that activates type-2 innate lymphoid cells Th2 immunity[27–29] (Fig. 4a, Supplementary Fig. 11), indicating an autocrine signaling loop.

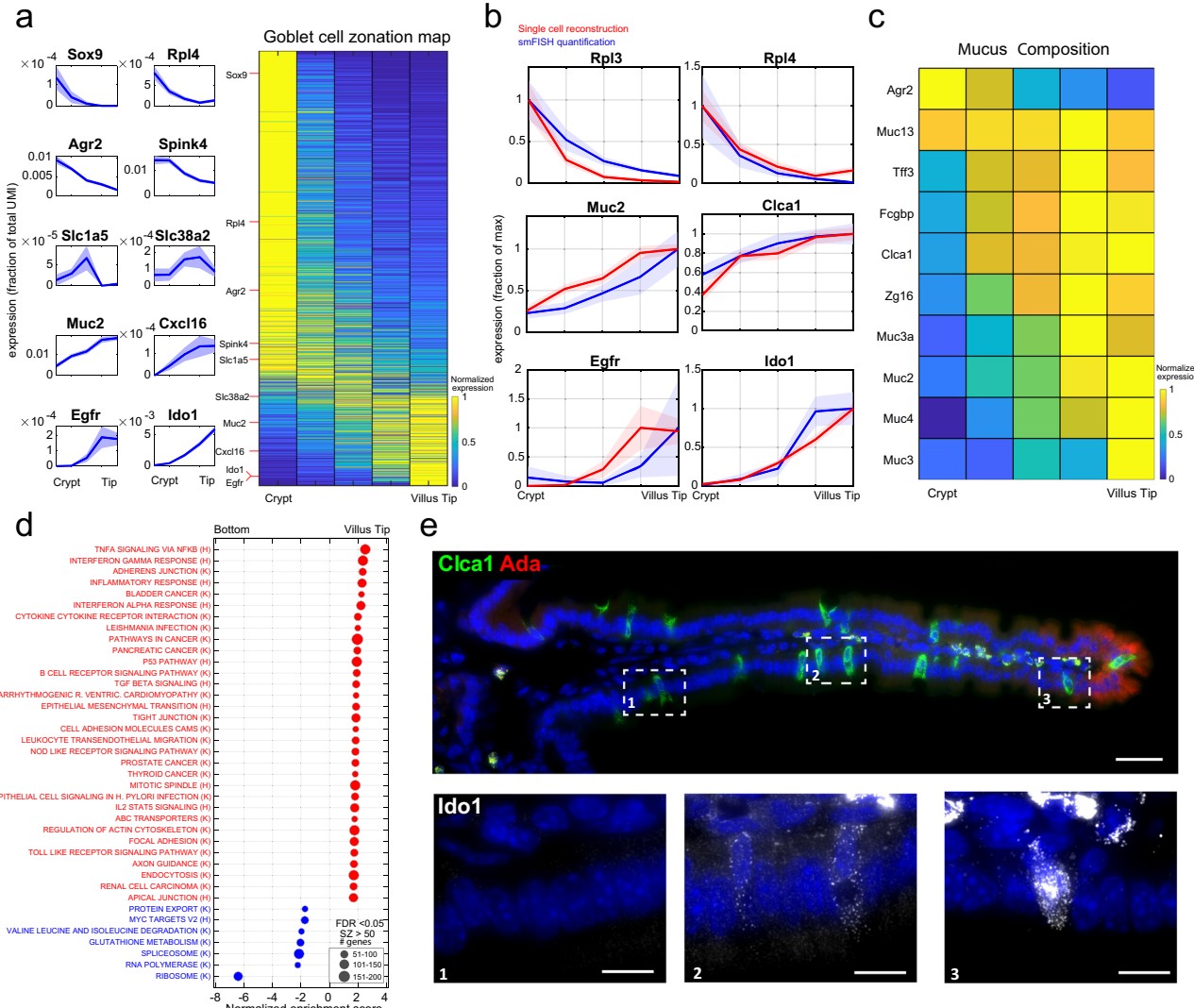

**Fig. 3 Spatial reconstruction of goblet cells. a** Reconstructed zonation profiles based on single goblet cells. Profiles are normalized to their maximum across zones. Plots on the left show zonation profiles of representative crypt (Sox9, Rpl4, Agr2, Spink4), mid-villus (Slc1a5, Slc38a2) and villus tip (Muc2, Cxcl16, Egfr, Ido1) genes. Light areas denote the SEM. **b** Validation of the reconstructed zonation profiles using smFISH. Blue line shows smFISH mean expression level, red line the reconstructed profile based on the single cell analysis. Light patches denote the SEM. Source data are provided as a Source data file. **c** Heatmap of zonation profiles of genes related to mucus composition. Profiles are normalized to their maximum across zones. **d** Gene set enrichment analysis[53] for hallmark (H) and KEGG (K) pathways (FDR < 0.05), tip enriched sets are in red, bottom enriched sets are in blue. **e** smFISH image of zonated genes Clca1 (green) and Ido1. Ada (red) marks the villus tip, blue the DAPI-stained nuclei. Bottom insets show Ido1 mRNAs (gray dots) increasing from the bottom (1) to middle (2) and tip (3) villus zones. Images representative of n = 10 villi (5/mouse). Scale bar, 50 μm in the stitched image and 15 μm in the insets.

A recent study identified two subsets of tuft cells termed "tuft1" and "tuft2" with distinct functions[17]. Tuft1 cells express neuronal-like genes, whereas tuft2 cells elevate immune programs, including the expression of *Ptgs1*, encoding the prostaglandin H2 synthase 1[17]. We found that tuft1-specific transcripts were zonated toward the bottom of the villus, whereas tuft2-specific transcripts were zonated toward the villus tip (Fig. 4e, Supplementary Figs. 11, 13).

**Enteroendocrine lineages have heterogeneous migration patterns.** Enteroendocrine cells are rare intestinal epithelial cells (~1%) that consist of diverse subtypes of hormone-secreting cells that are essential for physiological homeostasis. We next applied ClumpSeq to extract landmark genes for these cells. We extracted 656 enteroendocrine landmark genes from our 181 enteroendocrine-containing clumps (Supplementary Data 1) and used them to

infer the crypt-villus coordinates of single-sequenced enteroendocrine cells[12] (Supplementary Data 3, "Methods"). We found that around 35% of the highly expressed genes in enteroendocrine cells were significantly zonated (1838 out of 5243 genes expressed to levels above $5 \times 10^{-5}$ had q-value < 0.25). The zonated expression patterns conformed with previous observations of crypt-biased expression of *Gcg*, *Tac1*[30,31], and *Reg4*[32], and villus-biased expression of *Sct* and *Nts*[30] (Supplementary Figs. 14, 15).

A recent work analyzed the temporal expression programs of single enteroendocrine cells[12]. The study used a slowly-decaying fluorescent reporter, driven by *Neurog3*, a gene that is expressed in a pulse-like manner in the earliest crypt enteroendocrine progenitor, providing a time stamp for each enteroendocrine cell that enables grouping cells according to the time since their "birth". We argued that combining these temporal profiles with our spatial measurements could reveal the patterns of cell

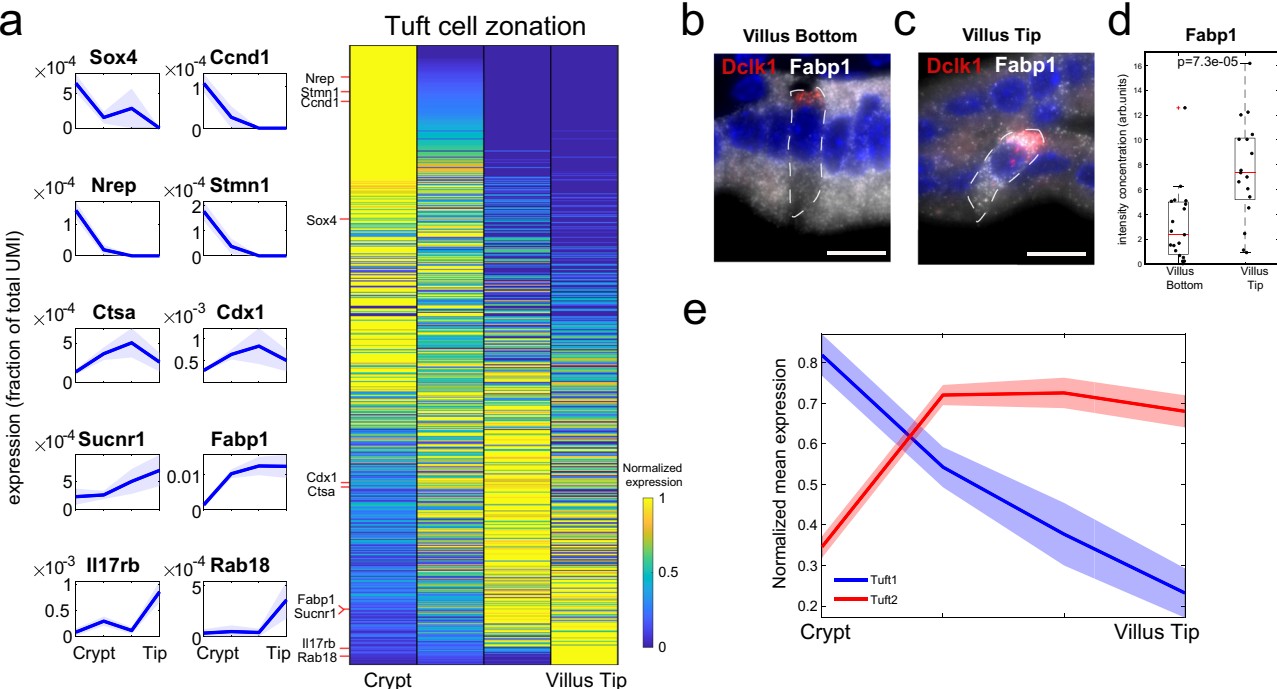

**Fig. 4 Spatial reconstruction of tuft cells. a** Reconstructed zonation profiles based on single tuft cells. Profiles are normalized to their maximum across zones. Plots on the left show zonation profiles of representative crypt (Sox4, Ccnd1, Nrep, Stmn1), mid-villus (Ctsa, Cdx1) and tip (Sucnr1, Fabp1, Il17rb, Rab18) genes. Light areas denote the SEM. **b, c** Representative smFISH images of the tuft zonated gene Fabp1 (gray) in a villus bottom (**b**) and a villus tip (**c**) tuft cell. Tuft cells were identified using Dclk1 (red). Scale bar 15 μm. **d** Quantification of Fabp1 smFISH experiment. *P* value was calculated by Mann–Whitney *U* test two-sided. *n* = 20 cells were examined over 2 mice. Red lines are medians, black boxes are 25–75 percentiles. Whiskers extend to the most extreme data point within 1.5× the interquartile range (IQR) from the box. Source data are provided as a Source data file. **e** Mean max-normalized zonation profiles for tuft1 and tuft2 genes. Light areas denote the SEM.

migration of different enteroendocrine cell types, identified by the expression of their characteristic hormones (Fig. 5a). More specifically, genes that are expressed in crypt cells that have an "early" time stamp, such as *Neurog3* and *Sox4*, are most probably transient genes expressed in cells that are migrating out of the crypt. Genes that are expressed in crypt cells with late time stamps, would indicate that the cells expressing them are stalling in the crypt. In contrast, genes that are expressed in villus cells with similarly late time stamps indicate faster migration of the expressing cells.

While the correlation between our inferred cryp-villus zone and the average time-stamp for different genes was significant ($R = 0.4$, $p < 10^{-10}$), different enteroendocrine genes exhibited distinct behavior in space-time (Fig. 5a). Cells expressing *Nts*, *Sct*, *Cck*, and *Gip* showed late time stamps and peaked in expression at the villus tip (Fig. 5a black-font genes, Supplementary Fig. 15a). Other genes, including *Reg4*[32], the X cell marker *Ghrl*, the EC cell marker *Tac1,* and the L cell markers *Pyy* and *Gcg* had intermediate-late time stamps, yet were confined to the crypt and villus bottom. This is in line with the previously shown transdifferentiation of Gcg+ cells into Cck+ and Nts+ cells, and of Tac1+ cells into Sct+ cells[30]. Notably, cells expressing the D cell marker *Sst*, encoding the hormone somatostatin, exhibited late time stamps, yet were spatially-confined to the crypt or lower villus zones (Fig. 5a, Supplementary Fig. 14). We used smFISH to demonstrate that D cells are indeed enriched in the crypt and lower villus zones (Fig. 5b, c, Fisher exact test $p = 3.8 \times 10^{-5}$). The discordant space-time profiles, with crypt retained expression of relatively late appearing genes, suggest that enteroendocrine cell types such as L-cells and D-cells have slower crypt-villus migration rates compared to the Cck+ I-cells and the Gip+ K-cells. We further used smFISH to validate that *Pyy* and *Afp*, two

enteroendocrine genes with late time stamps are enriched at the crypt and villus bottom (Supplementary Fig. 15b–e). Finally, we examined the differences in gene expression between D cells, predicted to migrate slowly, and Tac1+ EC cells and Gcg+ L cells, predicted to migrate more rapidly[30]. Gene set enrichment analysis[33] revealed that D cells were enriched in GO programs related to adhesion and migration (Supplementary Data 9). In particular, they highly expressed *Itbg5* and *Emp2*, genes involved in cell-matrix adhesion[34,35], *Mylk*, important for adhesion disassembly mechanism[36], and *Csf1* and its receptor *Csfr1*, implicated in the formation of unstable interactions[37] (Supplementary Fig. 16). The slower migration of D cells may thus be associated with more stable interactions with the extracellular matrix, or to more unstable interactions with neighboring epithelial cells. Future studies will explore whether D cells transdifferentiate into other enteroendocrine lineages at the villus tip or rather shed off at lower villus positions.

**Zone-specific interactions between epithelial and mesenchymal cells.** The zonated expression programs we identified using ClumpSeq suggested that secretory cells might be interacting with other zonated small intestinal cell types. Both enterocytes[7] and mesenchymal cells[38] were shown to exhibit strongly zonated gene expression signatures. We, therefore, sought to identify potential zone-dependent interactions. We analyzed a database of ligands and receptors[20] and identified pairs in which a ligand was enriched in one cell population and the matching receptor was enriched in another (Supplementary Data 10). We further examined ligand-receptor pairs enriched in either the crypt or bottom villus zones (Supplementary Fig. 17a) or in the villus tip zone (Supplementary Fig. 17b). Our analysis revealed classic interactions such as crypt telocyte *Rspo* genes and

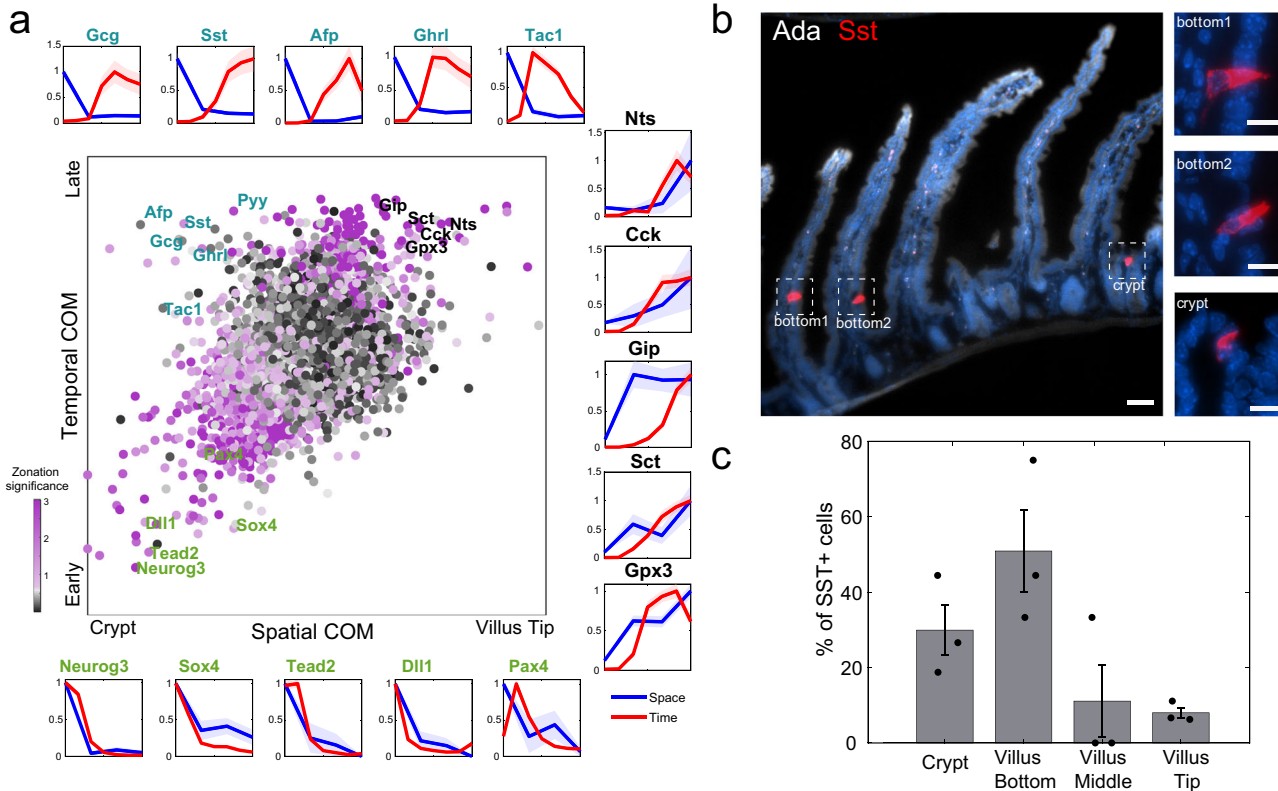

**Fig. 5 Migration pattern of eneteroendocrine lineages.** a Spatio-temporal analysis of enteroendocrine cells. Central plot: Scatter plot of the center of mass (COM) in our spatial zonation reconstruction vs. temporal COM based on single cell time stamps form Gehardt et al.[12] ("Methods"). Dots colored by −log10(spatial zonation q-value). Peripheral plots show reconstructed temporal (red) and spatial (blue) profiles for early crypt-confined genes (green fonts, bottom row), intermediate-late crypt/villus bottom-confined genes, expressed in cells that might be migrating more slowly (cyan fonts, top row) and late villus-localized genes, expressed in cells that might be migrating more rapidly (black fonts, right column). Light areas in plots denote the SEM. **b** Representative smFISH images of the enteroendocrine crypt-zonated gene Sst (red). Insets are blow-ups of the Sst+ cells at the crypt and bottom villus. Ada in gray marks the villi tips. Scale bars, 50 μm in the large image and 10 μm in the insets. **c** Quantification of Sst+ enteroendocrine D-cells in crypt and villus bottom, middle and tip over 3 mice. Fisher exact test (two-sided) for the frequencies of D-cells between the two lower zones and two upper zones p = 3.8 × 10^−5. Data are presented as mean values ± SD. Source data are provided as a Source data file.

crypt enterocyte Lgr5 and Lgr4, as well as signaling from tuft cells through the zonated ligands Inhbc, Dll3 and Ccl5 as well as sig-naling by zonated enteroendocrine cells (Supplementary Fig. 17a). The tip zone included the autocrine Il25-Il17rb circuit operating in tuft, as well as signaling from K enteroendocrine cells through the zonated Efna1, Ccl28, and Gip (Supplementary Fig. 17b).

## Discussion

ClumpSeq leverages the spatial information of the abundant cell types in a tissue to extract large sets of landmark genes for rare cell types. The ability to sort large clumps of up to 10 cells increases the capture rate of the rare cells, thus making it unne-cessary to use specialized cell-type specific surface markers for enrichment, markers that often do not exist. With larger clumps, the probability to contain more than two cell types increases, e.g. goblet cells and tuft cells in the same clump. While this was negligible in our study, the reconstruction algorithms can be readily adapted to take this into account by extracting landmark genes from a pool of secretory cell-type specific genes that are not expressed by both enterocytes and other secretory cells.

Recently, Gehart et al. uncovered distinct kinetics for different subtypes of enteroendocrine cells as well as different frequencies between crypts and villi[30]. Using ClumpSeq, we were able to

provide more detailed spatial information along the villus axis. For example, while Gehart et al. demonstrated that D cells have a late time-stamp and that Sst expression is higher in the villi compared to the crypts, we showed that along the villus, D cells are enriched at the villus bottom and that D cell frequency and Sst expression decreases toward the villus tip. Somatostatin inhibits the secretion of other hormones, such as Cck[39]. Moreover, using our ligand-recpetor analysis we found that most enteroendocrine cells express the somatostatin receptors (Supplementary Fig. 17a). Stalled D cells in the villus bottom might therefore serve to prevent the secretion of some of these hormones specifically at the lower villus zones.

While ClumpSeq directly provides the zonation profiles of cell-type specific genes, its real power of discovery emerges when using the two-step approach of first identifying a set of cell-type specific landmark genes to be used for reconstructing a genome-wide spatial single-cell atlas of the rare cell type (Fig. 1). This enables identifying zonated patterns of genes that are expressed in more than one cell type. An example is Fabp1, a gene that is highly abundant in enterocytes and peaks at the mid-villus zone, but is also zonated in tuft cells toward the villus tip. The expression of Fabp1 in clumps is dominated by the zonation of enterocytes, yet the tuft cell zonation emerges when recon-structing single sequenced tuft cells with the ClumpSeq-identified tuft cell landmark genes (Fig. 4b-d).

Our zonation results indicated that both goblet cells and tuft cells express immune-modulatory programs at the villus tip. We further showed that the neuronal-like tuft1 program is zonated toward the crypt and villus bottom, whereas the immune-related tuft2 program is zonated toward the tip. It will be important to utilize lineage tracing mouse models, such as in Beumer et al.[30] to examine whether these zonated cell states represent continuous trans-differentiation or rather distinct lineages that settle in different crypt-villus coordinates.

Recently, two powerful methods were developed for measuring spatial information with single-cell resolution, Slide-Seq[40] and High-definition spatial transcriptomics (HDST)[41]. Similarly to ClumpSeq, both methods need to be integrated with scRNAseq datasets in order to properly analyse the rare cells of interest. However, pre-requirements differ, as Slide-seq and HDST both work on fresh-frozen slides and require barcoding beads on a glass surface, while ClumpSeq uses freshly dissociated tissue with the clumps sorted in 384-well capture plates. Unlike Clumpseq, the spatial transcriptomics methods sequence thin tissue sections that often do not include complete cells, potentially posing challenges in the characterization of individual rare cell types.

ClumpSeq can be applied to diverse tissues and cell types, for example, the analysis of lung goblet and tuft cell diversity in spatially-distinct airways[42,43] and the zonation patterns of pancreatic endocrine cells along the radial islet axis[44]. ClumpSeq could also be adapted to assess the range-dependent effects of developmental organizers[45] and tumor signaling centers[46], thus expanding the toolbox of single cell biology beyond single cells and pairs.

## Methods

**Mice and tissue**. All mouse experiments were approved by the Institutional Animal Care and Use Committee of the Weizmann Institute of Science and performed in accordance with institutional guidelines (Protocol number 13000419-2). Experiments were conducted on 8–12 weeks old C57BL/6 mice, obtained from Envigo. Mice were housed at 4–5 per cage, maintained at a constant temperature of $22 \pm 2\,°C$ and humidity of $55 \pm 15\%$, exposed at all times to a 12 h light/12 h dark cycle and had access to food and water *ad libitum*. All experiments were performed on the same region of the Jejunum. Mice were sacrificed by cervical dislocation.

### Cell dissociation
*Clumps dissociation*. The Jejunum was harvested, flushed with cold 1× DPBS, laterally cut, and incubated for 20 min on ice in a 10 mM EDTA solution. Afterward, the tissue was cut into 1 cm pieces, moved in a pre-warmed 10 mM EDTA solution for 5 min and shaked vigorously at the end of the incubation time. Dissociated cells were collected and filtered through a 100 μm cell strainer. Cells were spun down at $300 \times g$ for 5 min at 4 °C. Pellet was resuspended and incubated for nuclear staining for 5 min at RT in a solution of DMEM + 10% FBS + 10 mM HEPES + Hoechst 33342 (15 μg ml$^{-1}$). To prevent the cells from pumping out the Hoechst dye, Reserpine (5 μM) was also added. Cells were resuspended in PBS and Alexa Fluo 488 Zombie Green (BioLegend) was added at a dilution of 1:500, to later enable the detection of viable cells by FACS. Cells were kept in a rotator in the dark at room temperature for 15 min. After spinning down ($500 \times g$. for 5 min at 4 °C), cells were resuspended in FACS buffer (2 mM EDTA, pH 8, and 0.5% BSA in 1× PBS) at a concentration of $10^6$ cells in 100 μl.

*Single cell isolation*. To obtain single cell suspension, rather than clumps, the tissue was incubated for 10 min on ice in a 10 mM EDTA solution, before to be cut in small pieces and moved for other 10 min in a pre-warmed 10 mM EDTA solution. The tissue was shaked vigorously every 2 min. Cells were filtered through a 70 μm cell strainer and spun down at $300 \times g$ for 5 min at 4 °C. Cells were resuspended in FACS buffer and stained with combination of APC-anti-Epcam (1:100, BioLegend, 118214) and PE/Cy7-anti-CD45 (1:1000, Biolengend, 103114) or APC-anti-Epcam and PE/Cy7-anti-CD24 (1:100, BioLegend, 101821). FcX blocking solution (BioLegend) was added at a dilution of 1:50.

### Clumps and single-cell sorting
Single cells and clumps were sorted with SORP-FACSAriaII machine with BD FACSDiva™ software (BD Biosciences) using a 100 μm nozzle. For clumps sorting, dead cells were excluded using the Zombie green staining and clumps were sorted based on Hoechst histogram (Fig. 1b, Supplementary Fig. 18a). For single cell sorting, dead cells were excluded on the basis of 500 ng/ml Dapi incorporation. Sorted cells were negative for CD45 and

positive for Epcam (Supplementary Fig. 18b). To enrich for enteroendocrine cells, cells were gated on CD45- Epcam+ CD24+. Since tuft cells express CD45[17], to enrich for those, cells were gated only on Epcam+ CD24+ (Supplementary Fig. 18c).

Cells and clumps were sorted into 384-well MARS-seq cell capture plates containing 2 μl of lysis solution and barcoded poly(T) reverse-transcription (RT) primers for single-cell RNA-seq. Barcoded single cell capture plates were prepared with a Bravo automated liquid handling platform (Agilent) as described previously[16]. Four empty wells were kept in each 384-well plate as a no-cell control during for data analysis. Immediately after sorting, each plate was spun down to ensure cell immersion into the lysis solution, snap frozen on dry ice and stored at −80 °C until processed.

**MARS-Seq library preparation**. Single cell libraries for both single cells and clumps were prepared, as described in Keren-Shaul et al.[16] Briefly, mRNA from cells sorted into MARS-Seq capture plates were barcoded and converted into cDNA by reverse transcription reaction and pooled using an automated pipeline. The pooled sample was cleaned using 0.9X SPRI beads and then linearly amplified by T7 in vitro transcription. The resulting RNA was fragmented and converted into sequencing-ready library by tagging the samples with pool barcodes and Illumina i7 barcode sequences during ligation, reverse transcription, and PCR. Each pool of cells was tested for library quality and concentration was assessed as described in Keren-Shaul et al.[16] Machine raw files were converted to fastq files using bcl2fastq package, to obtain the UMI counts. Using STAR (v.2.5.3a) reads were aligned to the mouse reference genome (GRCm38.84) using zUMI packge45[47] with the following flags that fit the barcode length and the library strandedness: -c 1-7, -m 8-15, -l 66, -B 1, -s 1, -p 16.

**scRNAseq data processing**. For each single cell or clump and for each gene we performed background subtraction. The background was calculated for each 384-well plate separately, as the mean gene expression in the four empty wells. After subtraction, negative values were set to zero. We used Seurat v3.2 package in R v3.6.1 and R studio v1.2.2019 to cluster the clumps and single cell RNAseq datasets, retaining only clumps or cells containing at least 200 genes. We used Seurat to regress out cell-cell variation driven by the fraction of mitochondrial genes. For clumps, we excluded clumps with over 30% mitochondrial genes. Clustering was based on PCA dimensionality reduction using the first 18 PCs, and a resolution value of 1.

For single cells, cells with either total UMI counts lower than 200 or higher than 7000 or total gene counts lower than 150 or higher than 1500 or mitochondrial content of over 40% were removed. Cell clustering was based on PCA dimensionality reduction using the first 25 PCs and a resolution value of 0.1. We used cell type-specific markers to interpret the single cell clusters: *Epcam* in the epithelial cells clusters, *Ptprc* in immune clusters, *Muc2* in the goblet cluster, *Dclk1* in the tuft cluster, *Chga* in the enteroendocrine cluster (Supplementary Fig. 7).

**ImageStream analysis**. Cells were imaged by an Imaging Flow Cytometer (ImageStreamX Mark II, AMNIS corp. - part of Luminex, TX, USA). Data were acquired using a ×40 lens, and lasers used were 405 nm (10 mW), 488 nm (100 mW), 642 nm (100 mW), and 785 nm (5 mW). Data were analyzed using the manufacturer's image analysis software IDEAS 6.2 (AMNIS corp.). Images were compensated for spectral overlap using single stained controls. Viable cells were first gated as negative for the dead cell marker Zombie-Green. To eliminate out-of-focus cells, cells were further gated using the Gradient RMS and contrast features (measures the sharpness quality of an image by detecting large changes of pixel values in the image). Then, cell were gated for single cells and cell clumps according to their area (in μm$^2$) and aspect ratio (the Minor Axis divided by the Major Axis of the best-fit ellipse). To distinguish between large cells and small clumps with similar size, the circularity feature was used (the degree of the mask's deviation from a circle, calculated as the average distance of the object boundary from its center divided by the variation of this distance)—high circularity was correlated with large cells rather than cell clumps. This was calculated using the Object cell mask (segments images to closely identify the area corresponding to the cell, by distinguishing it from the background), to better delineate cell morphology. To distinguish between pairs and larger clumps, objects were gated according to the area and aspect ratio (normalized for intensity) of the Hoechst staining. To validate that cell clumps contain more than one EpCAM positive cell, two features were calculated—the area of the EpCAM staining divided by the bright-field area, and the distance between the geometrical centers of the EpCAM staining and the bright-field image, using the Delta Centroid XY feature. Clumps with higher area ratio and lower distance were eventually chosen.

**Single molecule FISH and quantification**. Jejunum was harvested, flushed with cold 1× DPBS, laterally cut and then fixed in 4% formaldehyde for 3 h, incubated overnight with 30% sucrose in 4% formaldehyde and finally embedded in OCT in the form of swiss-rolls. 7 μm thick sections of fixed Jejunum were sectioned onto poly L-lysine coated coverslips and used for smFISH staining. Probe libraries were designed using the Stellaris FISH Probe Designer Software (Biosearch Technologies, Inc., Petaluma, CA). The intestinal sections were hybridized with smFISH

probe sets according to a previously published protocol[48]. Briefly, tissues were treated for 10 min with proteinase K (10 µg/ml Ambion AM2546) and washed twice with 2× SSC (Ambion AM9765). Tissues were incubated in wash buffer (20% Formamide Ambion AM9342, 2× SSC) for 5 min and mounted with the hybridization buffer (10% Dextran sulfate Sigma D8906, 20% Formamide, 1 mg/ml E.coli tRNA Sigma R1753, 2× SSC, 0.02% BSA Ambion AM2616, 2 mM Vanadyl-ribonucleoside complex NEB S1402S) mixed with 1:3000 dilution of probes. Hybridization mix was incubated with tissues overnight in a 30 °C incubator. SmFISH probe libraries (Supplementary Data 5) were coupled to Cy5, TMR or Alexa594. After the hybridization, tissues were washed with wash buffer containing 50 ng/ml DAPI for 30 min at 30 °C. DAPI (Sigma-Aldrich, D9542) was used as nuclear staining. All images were performed on a Nikon-Ti-E inverted fluorescence microscope using the NIS element software AR 5.11.01. All images were taken as scans extending from villus tip to crypt bottom using ×100 magnifications, hence several fields of view were stitched together to cover the whole crypt-villus unit. Stitching was performed with the fusion mode linear blending and default settings of the pairwise stitching plugin in Fiji[49].

Quantification of smFISH was done using ImageM[48]. Goblet cells were manually segmented based on *Muc2* or *Clca1* expression. Each transcript quantification was based on at least 5 entire villi from at least 2 mice. Tuft cells were manually segmented using *Dclk1* mRNA expression. Results were based on at least 20 cells from bottoms and tips of villi and from at least 2 mice. Enteroendocrine cells were identified based on the expression of the specific gene measured. Each villus was divided into bottom, middle and tip as follow: $Ada^+$ area is defined as "villus tip", the remaing villus length is divided equally in two zones, middle and bottom. Fisher exact test was calculated on the number of crypt-villus units with and without cells expressing the analyzed gene. mRNA concentration (number of mRNA per unit volume, for low abundance genes) or mRNA signal intensity (mean background-subtracted intensity in segmented unit, for high abundance genes) was computed per cell.

**Gene specificity analysis.** In order to find genes specific to the intestinal epithelial cell types, we comprised a table of mean expression of genes across cell types and the percentage of single cells of each cell type expressing each gene (Supplementary Data 6). To this end, we analyzed published scRNA-seq data sets[7,12,17,18], using cell type annotations by the papers' authors. Gene expression measurements (UMIs per gene) were normalized for each cell by the sum of its UMIs and then averaged across single cells by type. For enterocytes, we averaged cells from each villus zone using Moor et al. annotation[7]. The same single cell data source was also used for generating the crypt stem cells columns of Supplementary Data 6[18]. For other secretory cell types: goblet, enteroendocrine, tuft and paneth cells, we used the data from Fig. 1 of Haber et al.[17].

**Zonation reconstruction of clumps.** UMI counts table for all 5,297 clumps was exported from Seurat[50] and further clumps analysis was performed using MATLAB (version 2019a). 4,788 clumps with over 500 UMIs were retained and expression values per gene were calculated as UMIs per gene normalized for each clump by the total sum of its UMIs.

**Enterocyte landmark gene selection.** Enterocyte landmark genes for clumps zonation reconstruction were based on the enterocyte zonation table in Moor et al.[7]. There, gene zonation was reconstructed for the crypt and 6 villus zones (V1–V6) using single enterocytes. Candidate landmark genes were required to satisfy the following requirements: (1) Abundance—having a mean normalized expression across zones of $5 \times 10^{-4}$ or more. (2) Enterocyte specific: having mean expression in any enterocyte/stem-cell population higher than 10-fold the maximal mean expression in all secretory cell types and expressed in at least 10% of that enterocyte population. (3) Zonated—having at least 70% difference between maximal and minimal expression along the crypt-tip axis.

In order to select an informative set of landmark genes which includes crypt, mid-villus and villus tip markers, we calculated for all candidate landmark genes the Euclidean distances to "ideal" land mark profiles as follows: Ideal crypt landmark profile: expression value 1 in the crypt and 0 for all other zones; Ideal mid-villus land mark profile: expression value 1 in the middle of the villus (V3) and 0 for all other zones; Ideal tip land mark profile: expression value 1 in the tip and 0 for all other zones. Finally, three lists of enterocyte landmark genes were comprised: the crypt list with the 30 candidate landmarks with lowest distance from "ideal" profile 1, the mid-villus list with the 30 candidate landmarks with lowest distance from "ideal" profile 2 and the tip-villus list with the 30 candidate landmarks with lowest distance from "ideal" profile 3. If genes overlapped between the lists, they were assigned to the list of the "ideal" profile they were closest to. The selected landmark genes are shown in Supplementary Data 1.

**Assignment of clumps to zones.** Based on Supplementary Data 6, enterocyte specific genes were defined as genes for which the mean expression in any enterocyte/stem population was higher than 3-fold the maximal mean expression in all other secretory cell populations, and that were expressed in at least 10% of that enterocyte population. For comparability between clumps containing different numbers of enterocytes, these genes were internally normalized: their expression was divided by the sum total for all enterocyte specific genes in each clump. Note that the selected landmark genes are a subset of this group of enterocyte specific genes. 2% of clumps with lowest sums of enterocyte landmark gene expression were discarded, since they could not be reliably assigned to a zone. The remaining 4690 clumps were assigned a zone using the single cell enterocyte zonation table[7] as a spatial reference as follows.

The expression values of the enterocyte landmark genes in the spatial reference were normalized by dividing the expression of each gene by its maximal expression across zones. This resulted in a normalized landmark expression vector for each zone in the spatial reference. The expression of the enterocyte landmark genes in the clumps was also normalized by dividing the expression of each gene by its maximal expression across clumps. This resulted in a normalized landmark expression vector for each clump. Next, the correlations between the vector of landmark values for each clump were calculated with that of each of the zones. The clump was assigned to the zone it correlated most with. A clump-based zonation table was computed by averaging the expression values for each gene across all clumps in the zone. P values were calculated with the Kruskal–Wallis test (implemented in the MATLAB function kruskalwallis). q values were calculated using the Benjamini and Hochberg method (implemented in the MATLAB function mafdr), applied to all genes for which maximal expression across zones exceeded $5 \times 10^{-6}$ (Supplementary Data 2).

**Selection of cell type-specific classification markers.** Classification of clumps according to their contained cell type was performed separately for pairs and larger clumps due to differences in relative expression of genes stemming from clumps size. For cell type classification, we used Supplementary Data 6 to identify cell-type specific marker genes for secretory cells and enterocytes. For secretory cells, these included genes with mean normalized expression above $10^{-4}$, expressed in over 15% of the single cells and expressed at more than 4-fold higher levels than the maximal mean expression in all other epithelial cell types (we define this fold-change as specificity ratio). For each secretory type, all genes meeting these criteria were ordered by their specificity ratio in descending order and up to 50 first genes were selected as type markers. Enterocyte markers (used in Supplementary Fig. 3d) were selected similarly: genes expressed in at least 15% of enterocytes in any zone, with mean expression at least 4-fold greater in enterocytes than secretory cell types. The 50 genes with highest fold difference between enterocytes and secretory cells were selected. The list of cell-type classification markers appears in Supplementary Data 7.

**Geometric classification of clumps.** For each secretory cell type (goblet, tuft, enteroendocrine and Paneth), the expression levels of its classification markers were summed in each clump. These sums were converted to Z scores by subtracting the mean and dividing by the standard deviation across clumps. This process projected the clumps into a 4-dimensional space spanned by the sums of secretory cell type markers. We next performed principal component analysis on these shifted and scaled sums (implemented in the MATLAB function pca). This resulted in three principal components (PCs) that define a 3D position for each clump in PC space. For each PC, the median was subtracted in order to shift the origin of the PCs to the origin of axes. In PC space, the clumps were now arranged on four lines or rays emanating from the origin (Fig. 2o, Supplementary Fig. 3a). Clumps at the origin, where the sum of all secretory markers were low were enterocyte-containing clumps (Supplementary Fig. 3e). Clumps at the edge of each ray were the ones for which the sums of the distinct secretory type's markers were maximal. Intermediate clumps contained different contributions of the enterocyte transcriptome and the secretory cell transcriptome (Supplementary Fig. 3a). Larger clumps were closer to the origin, since the contributions of enterocytes, the major cell type, were higher in these clumps (Supplementary Fig. 3f).

We fitted a line to each of the secretory type rays. For the fit, we sorted clumps according to their distance from the origin and considered only clumps with distance above the 99 percentile. Fit was performed using a least square fit method implemented in a custom MATLAB script. Each ray was assigned to the secretory type, the markers of which peaked along it (Supplementary Fig. 3a). For each clump, the Euclidean distance from each of the rays was computed and Z scores for the four distances were calculated.

The farther from the origin of axes a clump was located, the higher it's sum of cell type-specific markers, and therefore the lower were the chances to miss-classify it. The region close to the origin contained clumps that were low in all cell type marker sums. There, a clump could be close to a particular ray at random. To minimize miss-classification we therefore sorted the clumps on each ray in descending order according to the distance from the origin, and included a fraction that matches the abundance of this cell type in the tissue. To estimate these abundances, we measured the proportions of each secretory cell type in both crypts and villi of the jejunum out of all cells (Supplementary Data 8). Measurements were performed by imaging the tuft cell marker *Dclk1*, the enteroendocrine cell marker *Chga* and the goblet cell marker *Clca1*. For Paneth cells, data was taken from Elmes M.J.[51]. For final thresholds, measured proportions for crypts and villi were multiplied by 2 for pairs and by 3 for larger clumps, to represent the higher probability to capture rare events in clumps. Secretory types were assigned only to

clumps far enough from the origin based on the above-mentioned cell type-specific thresholds, for crypt and villus clumps separately based on the respective threshold. Only clumps for which Z-score of distance from closest ray was below −1 were considered. All other clumps were classified as enterocyte-only clumps.

**Separation into clumps zonation tables per secretory type**. Clumps were separated by assigned secretory cell types, and the zone was assigned as previously described, based on enterocyte gene expression. For comparability between clumps containing different numbers of enterocytes, secretory cell type specific genes were internally normalized: their expression was divided by the sum of all secretory cell type specific genes in each clump. Secretory cell type specific genes, out of which a subset of secretory landmark genes were chosen (below), were defined as genes for which the mean expression in the secretory cell type was higher than 3-fold the maximal mean expression in all enterocyte populations and were expressed in at least 1% among cells in that secretory cell type. Calculation of zonation table proceeded as previously described for all clumps (Supplementary Data 2).

**Use of clumps for single cell zonation reconstructions**. For goblet, enteroendocrine and tuft cells, we used the clump-based zonation tables to find zonated, secretory-specific landmark genes. We then used the expression patterns of these landmark genes to assign single sequenced secretory cells to crypt-villus zones, grouped them and averaged their expression, thus obtaining zonation tables of all genes for each secretory cell type. The method we used for single cell reconstruction is similar for all three secretory types, with slight differences in landmark gene selection criteria and the spatial resolution of the reconstruction between goblet cells and the other secretory cell types. The reason for these differences stemmed from the substantially higher goblet cell abundance in the tissue and therefore in clumps, compared to other secretory cell. This enabled performing the reconstruction with finer spatial resolution for goblet cells. For all secretory types, single cell reconstruction consisted of the following steps, secretory type specific parameters and variations on this general method are detailed in the next sections:

1. Secretory cell type-specific genes were defined as genes for which mean expression in the secretory cell type was higher than 3-fold the maximal mean expression in all enterocyte populations, and that were also expressed in at least 1% of secretory cells.
2. For comparability between clumps containing different numbers of cells, these genes were internally normalized: their expression in each clump was divided by the sum of all secretory cell type-specific genes in that clump.
3. Out of the secretory specific genes in step 1, two groups of zonated landmark genes were selected: crypt landmark genes, which are zonated toward the crypt and villus-tip landmark genes, zonated toward the villus tip. See sections below for detailed description of landmark gene selection criteria. For each clump of this secretory cell type, the sum of the normalized expression of the crypt landmark genes (denoted X) and of the tip landmark genes (denoted Y) was calculated. These sums were than used to calculate a unit-less spatial coordinate: (1) $\eta = Y/(X + Y)$ for each clump. This yielded a distribution of $\eta$ values for each zone in secretory cell-containing clumps.
4. For each single secretory cell used for reconstruction, the spatial coordinate $\eta$ was calculated as in step 3.
5. In order to assign each single cell to a zone based on it's $\eta$, $\eta$ limits for the zones were calculated using an optimization method—reconstruction was performed using a wide range of possible $\eta$ limits options (see step 6 for details). The set of $\eta$ limits that yielded zonation profiles of secretory specific genes which best fit the clumps profiles was selected. Specifically, for each possible set of $\eta$ limits:
   a. We performed single cell zonation reconstruction as described in steps 7–8
   b. For each secretory specific gene (defined in step 1), the zonation profile in the current reconstruction was compared to the zonation profile in clumps: the Euclidean distance between the two profiles was calculated after both were normalized by their maximal values.
   c. The median over genes of this distance was calculated and denoted as $Med_{Euc.}$
   d. The set of $\eta$ limits yielding the smallest $Med_{Euc}$ was selected as optimal.
6. All $\eta$ limit sets that were considered in the optimization described in step 5 were calculated as follows:
   a. The lowest possible upper bound on $\eta$ for crypt was set to the median of $\eta$ values of crypt clumps—calculated in step 3. We denote this number as $\eta_{min}$
   b. The resolution of $\eta$ optimization denoted $D_\eta$, was determined.
   c. A vector of all considered $\eta$ limit values, $\eta_{Vec}$, was created: a regularly-spaced vector starting at $\eta_{min}$ and ending in 1, using $D_\eta$ as the increment between elements. $\eta_{Vec}$ elements were therefore: $[\eta_{min}, \eta_{min} + D_\eta, \eta_{min} + 2* D_\eta,…, \eta_{min} + m* D_\eta]$ where $m = (1 −\eta_{min})/D_\eta)$.
   d. All possible combinations of the elements of $\eta_{Vec}$ taken $N_{zones}−1$ ($N_{zones}$ is the number of desired zones for the reconstruction) at a time were calculated. Each such combination is an optional set of $\eta$ limits,

with a zero added in the beginning and one appended at the end. For example, if the reconstruction is to be done for 4 zones, each such optional set of $\eta$ limits would be: 0, $\eta_{Vec1}$, $\eta_{Vec2}$, $\eta_{Vec3}$, 1. With $\eta_{Vec1}$, $\eta_{Vec2}$, $\eta_{Vec3}$ being one of the combinations of $\eta_{Vec}$ values- such that $\eta_{Vec1 <} \eta_{Vec2 <} \eta_{Vec3}$.
   e. All sets of $\eta$ limits which yielded less than 10 single cells in some zone were discarded and not considered in the optimization described in step 5.
7. Each single cell was assigned to a zone based on the optimal $\eta$ limits selected as described in step 5.
8. The expression values of all the single secretory cells in each zone were averaged for each gene, to obtain the zonation table of genes in the secretory cells.
9. $P$ values for zonation per gene were calculated with the Kruskal–Wallis test (implemented in the MATLAB function kruskalwallis). $q$ values were calculated using the Benjamini and Hochberg method (implemented in the MATLAB function mafdr), for all genes for which expression exceeded $5 \times 10^{-6}$.

**Zonation reconstruction of single goblet cells**. Single cells used for zonation reconstruction were from scRNA-seq experiments on intestinal cells, performed using the MARS-seq protocol (Supplementary Fig. 7), see scRNA-seq section for details. Goblet cells were detected based on Seurat clustering[50].

We defined the Center Of Mass (COM) of a gene's spatial expression profile across the zones 1,2,..N with respective expression values per zone of $E_1,E_2,…E_N$ as:

$$COM = \sum_{i=1}^{N} E_i*i/ \sum_{i=1}^{N} E_i \qquad (1)$$

Goblet cell specific landmark genes based on clumps data (steps 3–4 in the previous section) were selected based on the following criteria:

- Maximal expression across zones in clumps zonation Supplementary Data 2 $> = 5 \times 10^{-5}$.
- Crypt markers: COM $<= 3$, expressed in at least 2 clumps in the crypt.
- Tip markers: COM $> = 4.7$, expressed in at least 2 clumps in the tip most zone.

This resulted in 309 crypt markers and 62 tip markers listed in Supplementary Data 1. Single cell reconstruction was performed with 5 zones. Resolution parameter for optimization- $D_\eta$ (step 6b in the previous section) was set to 0.05.

**Zonation reconstruction of single enteroendocrine cells**. Single cells used for zonation reconstruction were taken from Gehart[12]. UMI count table for the single cells was downloaded from Gehart et al.[12] (GEO: GSE113561) and parsed in MATLAB. Expression values per cell were normalized by dividing by the overall sum of UMI for each cell. Cells marked as excluded in the metadata supplied by the authors were removed. Enteroendocrine specific landmark genes based on clumps data (steps 3–4 in previous section) were selected based on the following criteria:

- Maximal expression in clumps zonation Supplementary Data 2 $> = 5 \times 10^{-5}$.
- Crypt markers: COM $<= 1.8$, expressed in at least 2 clumps in the crypt.
- Tip markers: COM $> = 4.2$, expressed in at least 2 clumps in the tip most zone.

This resulted in 636 crypt markers and 20 tip markers listed in Supplementary Data 1. Single cell reconstruction was performed with 4 zones. Resolution parameter for optimization- $D_\eta$ (step 6b in the previous section) was set to 0.02.

To obtain the temporal profile of enteroendocrine gene expression (Fig. 5a), time stamps per cell, which were available in GEO: GSE113561 were used to equally partition cells into 7 temporal bins, assigning each cell to a distinct temporal zone. Within each temporal bin, gene expression was averaged over cells in that bin creating the temporal expression table. $P$ values for temporal profiles per gene were calculated with the Kruskal–Wallis test (implemented in the MATLAB function kruskalwallis). $q$ values were calculated using the Benjamini and Hochberg method (implemented in the MATLAB function mafdr).

**Zonation reconstruction of single tuft cells**. Our single cell MARS-seq protocol yielded mainly villus tuft cells. We, therefore, combined our cells with tuft cells from Fig. 1 of Haber et al.[17]. Zonation reconstruction was performed separately for these two single cell datasets and consequently merged. Tuft cell-specific landmark genes based on clumps data (steps 3-4 in previous section) were selected based on the following criteria:

- Maximal expression in clumps zonation Supplementary Data 2 $> = 5 \times 10^{-5}$.
- Crypt markers: COM $< = 1.8$, expressed in at least 2 clumps in the crypt.
- Tip markers: COM $> = 4.2$, expressed in at least 2 clumps in the tip most zone.

This resulted in 323 crypt markers and 29 tip markers listed in Supplementary Data 1. Single cell reconstruction was performed with 4 zones. Resolution parameter for optimization- $D_\eta$ (step 6b in previous section) was set at 0.02. The merged zonation table was calculated as the weighted mean of the two zonation tables derived separately from the two datasets with the weights reflecting the relative contribution of each data set per zone in terms of amount of expressing cells.

The details of the calculation are as follows—for each possible set of $\eta$ limits:

1. The single cell reconstruction using only cells from our data was computed.
2. The single cell reconstruction using only cells from Haber et al.[17] was computed.
3. For each gene, the mutual zones between data sets were identified. Mutual zones were defined as zones in which both data sets had 5 or more cells expressing the gene above the expression threshold, set at $5 \times 10^{-6}$.
    i. If there were no mutual zones-and the gene was expressed only in one dataset, the expression in that data set was retained for the merged zonation table.
    ii. If both data sets had cells expressing the gene, but without sufficient overlap in a single zone the gene was excluded from the merged zonation table.
    iii. If there were several mutual zones, the mutual zone in which the data sets had the most similar amount of expressing cells was selected.
4. For each gene, the zonation expression profile in each data set was normalized by the value in the mutual zone.
5. These two scaled profiles were averaged with weights per zone. The weights per dataset per zone were the amount of cells expressing the gene above the expression threshold ($5 \times 10^{-6}$).
6. Averaged zonation profiles for each gene were re-normalized by dividing by the maximal value, and re-scaled by multiplying the normalized profiles by the maximal expression level between the two separate zonation tables.

For the merged zonation table, the p values were calculated per gene as the minimal $p$ value between the two separate reconstructions. $q$ values were calculated using the Benjamini and Hochberg method (implemented in the MATLAB function mafdr).

**Validation of single cell reconstructions with clumps**. In order to further validate the single cell reconstructions, we computed the correlation of centers of mass (COM) between single cell and clump-based reconstructions. For this puropose, we coarse-grained the 7-zone clumps zonation tables into the same number of zones as the respective single cell reconstruction. These coarse grained zonation tables were calculated the same way as described for clump to zone assignment, with one difference: the spatial ref.[7] zonation table was linearly interpolated for the smaller number of equally spaced zones prior to reconstruction.

For the validation, we chose genes that were not used as landmarks for single-cell reconstruction and were both highly expressed (above $10^{-5}$) and secretory specific (secretory specific criteria described in step 1 of the single cell reconstruction algorithm). We further limited ourselves to genes for which the SEM of reconstruction was small enough for both single cells and clumps based reconstructions (below 0.4 for goblet and below 0.5 for tuft). This yielded a similar quantity of genes for goblet and tuft: 30 and 28, respectively.

For Enteroendocrine cells, such validation was infeasible, due to the various enteroendocrine cell sub-types and their relatively sparse representation in clumps. Instead, single cell reconstruction of spatial zonation was validated against the temporal gene expression patterns derived from the same single cells in Gehart et al.[12], resulting in high correlation ($R = 0.4$, $p < 10^{-10}$).

**Robustness analysis**. In order to assess the robustness of the zonation tables to parameter choices, we reconstructed the single cell zonation tables with modified parameter values and examined the change in the correlation of the zonation profiles centers of masses (COM) between the original and perturbed zonation tables as a function of the size of introduced parameter perturbation. To this end, we performed the following for all parameters pertaining to clump secretory type assignment and subsequent single cell reconstructions (Supplementary Data 11): we selected ~75 linearly spaced values for each parameter within the 0.5–1.5 fold range. For some parameters, only integer values could be used. For other parameters, not the entire 0.5–1.5 fold range yielded reconstructions due to absence of marker or landmark genes fitting the criteria for these values. For each parameter, we then generated single cell zonation tables for all the 75 different parameter values. In order to compare these reconstructions to the original zonation table, we calculated the Spearman correlation coefficient of the COMs for the expressed genes (above $5 \times 10^{-6}$), between each reconstruction and the original. We denote this quantity for parameter i and perturbed value j – $P_{ij}$. We then plotted for each parameter i, the values $P_{ij}$ for j = 1:75. The plotted curves (Supplementary Figs. 4a, 5a, 6a) were smoothened by applying a moving median filter with window size 10. Due to smoothing, the point (0,1) which indicates that for 0 distance between the perturbed and original parameter, the Spearman correlation $P_{ij}$ is 1, was sometimes shifted. We, therefore, reintroduced this point after applying the moving median filter.

In order to visualize the differences in the zonation profiles of individual genes between the original and perturbed reconstruction when $P_{ij}$ declines, we selected for each cell type a representative example of a parameter choice that yielded low correlation ($P_{ij}$) and plotted the original and the perturbed reconstructed zonation profiles for 10 selected genes: 5 crypt genes and 5 tip genes (Supplementary Figs. 4b, 5b, 6b). The genes shown for these examples were selected as follows: several gene types were excluded (exclusion criteria below), the remaining genes were sorted by COM and top 5 (tip) and bottom 5 (crypt) were auto-selected.

Genes were excluded based on the following criteria, considering the original reconstruction:

1. landmark genes used to generate the original reconstructions.
2. low confidence genes—$q$ value above 0.05.
3. not highly zonated genes—less than 2 fold dynamic range.
4. lowly expressed genes—with maximal normalized expression below $5 \times 10^{-5}$.

**Ligand-receptor analysis**. Ligand-receptor analysis was performed similar to Bahar-Halpern et al.[38]. We performed the analysis between and within zonated epithelial and mesenchymal cell types. We used the zonated secretory cell expression reconstructed with Clumpseq and previously published datasets for zonated enterocytes and mesenchymal cells[7,38]. A list of ligand-receptor pairs was extracted from Ramilowski et al.[20] (697 unique ligands and 688 unique receptors). For each gene g and each cluster c we calculated the average expression $x_g^c$. We then computed a Z-score, $Z_g^c$, representing the enrichment of each ligand and receptor in each cell type:

$$Z_g^c = \frac{x_g^c - \mathrm{mean}(x_g^c)}{\mathrm{std}(x_g^c)} \qquad (2)$$

where the mean and standard deviations were compute over all cell types. We next defined an interaction score as:

$$Z_{\mathrm{interaction}} = \sqrt{\left(Z_L^{C1}\right)^2 + \left(Z_R^{C2}\right)^2} \qquad (3)$$

where $Z_L^{C1}$ is the ligand Zscore for cell type C1, and $Z_R^{C2}$ is the receptor Zscore for cell type C2. The resulting list of interactions was filtered per fraction of cells expressing either the ligand or the receptor (>0.05) and the $Z_{\mathrm{interaction}}$ was above 2.

Cytoscape[52] was used to visualize bottom and tip interactions. We selected only ligands and receptors with an average expression of either the ligand or the receptor above $2 \times 10^{-5}$ and $Z_{\mathrm{interaction}}$ higher than 5. For interactions that occur at the crypt or villus bottom zones (Supplementary Fig. 17a), we considered EC, D, L, X and N enteroendocrine cells and crypt-villus bottom goblet, tuft, enterocytes and telocytes. For interactions at the villus tip (Supplementary Fig. 17b), EC, I, L and N enteroendocrine cells and villus tip goblet, tuft, enterocytes and telocytes.

**Reporting summary**. Further information on research design is available in the Nature Research Reporting Summary linked to this article.

## Data availability
Data generated in this study hve been deposited in Gene Expression Omnibus with the accession code: "GSE154714". Single cell dataset also includes epithelial cells from "GSE134479"[38]. Enteroendocrine single cell dataset was acquired from NCBI GEO dataset browser, with accessions code: "GSE113561"[12]. All other relevant data supporting the key findings of this study are available within the article and its Supplementary Information files or from the corresponding author upon reasonable request. Source data are provided with this paper.

## Code availability
All codes used in this study will be available upon request. All codes are uploaded on Zenodo (https://doi.org/10.5281/zenodo.4561515).

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

## Acknowledgements

We thank all members of our lab for valuable comments. S.I. is supported by the Wolfson Family Charitable Trust, the Edmond de Rothschild Foundations, the Fannie Sherr Fund, the Dr. Beth Rom-Rymer Stem Cell Research Fund, the Helen and Martin Kimmel Institute for Stem Cell Research grant, the Israel Science Foundation grant No. 1486/16, the Broad Institute-Israel Science Foundation grant No. 2615/18, the European Research Council (ERC) under the European Union's Horizon 2020 research and innovation programme grant No. 768956, the Chan Zuckerberg Initiative grant No. CZF2019-002434, the Bert L. and N. Kuggie Vallee Foundation and the Howard Hughes Medical Institute (HHMI) international research scholar award.

## Author contributions

R.M., K.B.H., and Z.P. performed the experiments, I.A. and S.I performed the data analysis, R.M. and I.A. contributed to project design. S.I., R.M., and I.A. wrote the manuscript. All of the authors discussed the results and commented on the manuscript.

## Competing interests

The authors declare no competing interests.
