## [Peer Review File · Nature Communications]

Reviewers' Comments:

Reviewer #1:

Remarks to the Author:

The manuscript by Manco et al. describes Clumpseq approach. The authors performed RNA-sequencing of 2-10 intestinal epithelial cells. Based on single-cell RNA sequencing data from rare cell types, including goblet, entero-endocrine and tuft cells and Laser Capture Microdissection single-cell transcriptome data from enterocytes, the authors assigned positional identities of the rare cell types to various regions of the small intestinal epithelium. Those included crypts, lower villi, mid-villi and villi tips. They found that goblet cells expressing ribosomal proteins or transcription factor Sox9 were located within the crypts. In contrast, the expression of mucins was higher in the goblets located in the upper villi region compared to the lower villi. Similar, tuft cells expressing Cyclin D1 or Stmn, the markers of proliferating cells, were located in the crypts, whereas tuft cells expressing Fabp1 were located in the upper villi. Furthermore, using this approach, the authors confirmed regional identity and time of differentiation for EECs. Finally, they found that both goblet and tuft cells located at the upper villi express genes implicated in innate immune response at higher levels compared to the cells located in the crypts or lower villi, suggesting that the maturation of goblet and tuft cells accompanied by their "immune-specialization".

The presented study is purely descriptive. The concept is not novel. I agree, that the technique could be used for pilot studies, whether it is named Clumpseq or ultra-low RNA-seq, but ultimately one needs scRNA-seq data to either identify rare cell types or to learn about landmarks from the surrounding cells. I would expect to have the landmarks within the rare cell types. Less differentiated progenitors should be within the crypts, having a higher proportion of transcripts for ribosomal and cell cycle promoting genes, whereas more differentiated cells should express more specialized markers. Furthermore, either RNA in situ hybridization or antibody staining are still required to confirm the spatial localisation of the rare cells.

In the abstract, the authors claim: "We uncover immune-modulatory programs in villus tip goblet and tuft cells and heterogeneous migration patterns of enteroendocrine cells." Describing the differential expression of some genes implicated in immune response or targets of interferon signalling does not mean discovering immune-modulatory programs. Moreover, heterogeneous migration patterns of enteroendocrine cells were already discussed by Gehart and colleagues, 2019.

Other points:

1. The authors should check for the list of "ligands" in Suppl. Fig. 4. Hspa1a (encoding for heat shock protein) and Rps19 (encodes for ribosomal protein) are not ligands.
2. In Beumer et al., PYY+ cells were found mostly in the villi. Here, the cells expressing Pyy at the highest level are in the crypts (Suppl. Fig 6). How do the authors explain such discrepancy?

Reviewer #2:

Remarks to the Author:

The present study by Manco et al. introduces ClumpSeq a method for reconstructing spatial zonation of rare cell types in the intestinal epithelium. ClumpSeq involves sequencing of partially digested tissue clumps comprising ~2-10 cells each. The authors had previously inferred spatial zonation of enterocytes representing the most abundant intestinal epithelial cell type (Moor et al., 2018, Cell) by combining single-cell RNA-seq and in situ quantification of landmark genes using smFISH. The enterocyte zonation profiles from this study were now used as a reference, in order to assign each sequenced clump to a spatial zone. By utilizing secretory cell type-specific genes, zonation coordinates of clumps containing a given secretory cell types were used to match single-cell RNA-seq data of the same secretory cell type, permitting the inference of genome-wide spatial zonation profiles for this cell type.

ClumpSeq was applied to investigate patterns of transcriptome zonation and enrichment of functional pathways in distinct zones along the crypt-villus axis.

This is very nice study presenting a robust approach for resolving cell type architecture in spatially organized systems.

The experimental data are of very good quality, and the computational analysis pipeline is solid,

although it involves some ad hoc parameter choices, which seem somewhat arbitrary. The core idea of ClumpSeq is not entirely novel. The authors applied a similar strategy in the past to derive zonation of endothelial cells in the liver (Halpern et al., 2018, Nature Biotechnology), where they sequenced pairs of hepatocytes and endothelial cells to derive zonation of the latter based on hepatocyte zoned landmark genes inferred from smFISH (Halpern et al., 2017, Nature). The major novel aspect of the current approach is the computational deconvolution strategy to infer zonation of rare cell types from ClumpSeq, and this approach could be very useful for revealing localization of rare cell types in other spatially organized tissues. With this method the authors report some interesting and novel findings on spatial zonation of goblet, tuft, and enteroendocrine cells.

The manuscript is concise and well written and refers to the relevant single-cell studies on intestinal epithelial cell types.

I have only a few concerns and suggestions for additional analyses to strengthen the paper:

1. The algorithm involves a number of assumptions and ad hoc parameter choices, e.g., for the definition of cell type specific genes or landmark gene. The authors should include a computational analysis to demonstrate the robustness of the inferred zonation coordinates with regard to the choice of these parameters. Another example is the factor used to scale the abundance of secretory cell types in order to select the fraction of clumps along the rays in the PCA to be included into the analysis. This factor was chosen as 2 for pairs, and 3 for large clumps. The latter choice seems quite coarse as these clumps could comprise any number of cells up to ~ 10 , and thus the expected probability of finding a secretory cell in a clump is expected to vary accordingly. Hence, these quantities should also be subject to a robustness analysis. Since the authors are able to record index data with their sequencing approach, this factor could be modeled more systematically given the number of cells in each clump estimated from DNA staining. My worry is that a too conservative cutoff potentially eliminates clumps with less pronounced expression of cell type specific genes, which could be localized to specific zones, e.g., progenitor stages, naively expected to be localized preferentially to the crypt bottom.
2. The authors validate the spatial reconstruction of goblet cells by smFISH for a number of markers with different zonation patterns, which very nicely shows the quality of the ClumpSeq inference. However, for enteroendocrine and tuft cells they only include smFISH validation for one or two genes, respectively. Please include a few more genes with complementary zonation patterns for each cell type.
3. Is it possible to reconstruct a differentiation program of goblet cells from their zonation profile? Naively, I would expect that maturation concurs with migration upward along the crypt-villus axis, and thus the differentiation stage should be a correlate of the zonation coordinate. This should be tested for the other secretory cell types as well, beyond the timestamp analysis presented for enteroendocrine cells.
4. It would be helpful to include a plot showing the expected frequency of each rare secretory cell type along the crypt-villus axis derived from the ClumpSeq data. This has been somewhat addressed for some of the enteroendocrine cells, but a summary figure showing the frequency of all secretory cell types as a function of the crypt-villus coordinate derived from the enterocyte profiles would be informative.
5. An important goal of spatial gene expression analysis is the inference of molecular interactions between different cell types. Is there spatial co-localization of specific zoned sub-types of different populations, and is it possible to infer *in silico*, if these sub-types are interacting through specific molecular pathways? For example, are there ligand-receptor pairs suggesting the interaction of enteroendocrine cells confined to the lower crypt with crypt-bottom enterocytes? Could these interactions be involved in controlling the local maturation of one or the other cell type?
6. The authors argue based on the observed anticorrelation of crypt and villus tip enterocyte markers, as well as based on the expression of crypt enterocyte markers and Paneth cell markers, that clumps are a result of incomplete digestion rather than forming from individual cells in solution. However, although these data indicate that the major contribution comes from actual tissue clumps, the "background" contribution from artificial clumps emerging after complete dissociation from single cells should be analyzed. This should be done experimentally by quantifying the number of clumps reshaping in solution after longer digestion using FACS.

Reviewer #3:

Remarks to the Author:

In the submitted manuscript Rita Manco, Inna Averbukh and colleagues describe a new approach – ClumpSeq - to catalog rare cell in the small intestine. ClumpSeq is an interesting technique, which together with additional approaches can be used to infer the spatial situation of different cell types in the tissue. Using this method, the authors succeeded to capture rare cells as goblet cells, enteroendocrine and tuft cell and distinctively infer their gene expression profiles specific to the different zonation. The study is well performed, claims are supported by data. The manuscript may represent an important method/resource paper for scientists interesting in spatial and possibly also functional characterization of rare cell types at single cell level. Nevertheless, there are some issues that should be addressed in a revision. Majority of them do not point to weak or critical points but represent questions that arose during the reading of the manuscript as result of the curiosity. Hopefully addressing them may be helpful also for the authors.

- Combination of ClumpSeq with LCM and conventional scRNA-seq provides a more detailed spatially corrected transcriptomic profile that could help predict/hypothesize more complex cellular features. An intriguing concept that could be addressed in this work is to try and provide more explanation regarding cellular dynamics. For instance, can the authors analyze whether villus-tip differentiated cells (goblet, tuft) are derived from the same cell types previously located at the villus bottom/crypt or there are different progenitors that can give rise to different bottom and top cells? Do the tip cells differentiate in the tip or they can be dynamic while migrating towards the top, acquiring different identities? Can some cellular trajectories be charted for rare epithelial cells describing the development of distinct spatial sub-types of rare cells?

- Can be data acquired in this manuscript somehow integrated with previous datasets determining intestinal mesenchymal cells (McCarthy et al., 2020) or zoned intestinal mesenchyme generated previously by Itzkovitz lab (Bahar Halpern et al. 2020). Such an integratory approach may reveal a specific, rare/secretory cells-mesenchymal interactions. For example, SuppFig.4b indicates high expression of *Pdgfa* by villus-tip goblet cell. At the villus *Pdgfra* mesenchymal cells are located (McCarthy et al., 2020). Is it just a co-occurrence? Or based on Figure 3e, it seems that *Ido1* is also expressed in mesenchymal cells that co-express *Cla1*. Can authors comment this?

- What exactly does it mean “immune-specialization of goblet cells at the tip of the villus” except of enhanced IFN-response? Is it such a response specific to goblet cells at the villus-tip? Or other villus tip cells (incl. enterocytes) are also affected? Recent data indicated possible impact of IFN γ on cellular differentiation in the crypt (Biton et al., 2018), including restriction of secretory cell differentiation (Sato et al., 2020). Are any of these mechanisms relevant also for tip-villus goblet cells?

- Can the authors identify progenitors vs. newly differentiated cells in the crypt? For instance, in page 8 the authors state that “Tuft cells at the crypt expressed the transcription factor *Sox4*”. Are these Tuft cells progenitors or differentiated Tuft cells? Based on Figure 4a, is it correct that they are still proliferative, based on expression of *Ccnd1*? What about other rare cell types?

- Are there any transcriptional differences between the proposed slow vs. fast migrating enteroendocrine cells that could explain their different behavior? Perhaps any gene signature featuring migratory characteristics? In addition, staining for villus top enteroendocrine cells could also be interesting to add. Can authors speculate what could be the functional implication of these differences?

- It would be helpful to add to Figure 4b/c, a staining with lower magnification to appreciate the expression pattern of *Fabp1* along the crypt/villus axis.

- It might be worth to compare/discuss ClumpSeq (pipeline) with other single cell approaches allowing spatial transcriptomics as for example Slide-seq (Rodrigues et al., 2019). Maybe some table indicating pre-requirements/resolution/sensitivity etc. would be nice.

We thank you and the reviewers for the excellent and constructive comments on our manuscript. We have now addressed all of the comments with extensive new experiments and analyses, as detailed below. We believe these have significantly improved our study.

Reviewer #1 (Remarks to the Author):

The manuscript by Manco et al. describes Clumpseq approach. The authors performed RNA-sequencing of 2-10 intestinal epithelial cells. Based on single-cell RNA sequencing data from rare cell types, including goblet, entero-endocrine and tuft cells and Laser Capture Microdissection single-cell transcriptome data from enterocytes, the authors assigned positional identities of the rare cell types to various regions of the small intestinal epithelium. Those included crypts, lower villi, mid-villi and villi tips. They found that goblet cells expressing ribosomal proteins or transcription factor Sox9 were located within the crypts. In contrast, the expression of mucins was higher in the goblets located in the upper villi region compared to the lower villi. Similar, tuft cells expressing Cyclin D1 or Stmn, the markers of proliferating cells, were located in the crypts, whereas tuft cells expressing Fabp1 were located in the upper villi. Furthermore, using this approach, the authors confirmed regional identity and time of differentiation for EECs. Finally, they found that both goblet and tuft cells located at the upper villi express genes implicated in innate immune response at higher levels compared to the cells located in the crypts or lower villi, suggesting that the maturation of goblet and tuft cells accompanied by their “immune-specialization”.

1. The presented study is purely descriptive. The concept is not novel. I agree, that the technique could be used for pilot studies, whether it is named Clumpseq or ultra-low RNA-seq, but ultimately one needs scRNA-seq data to either identify rare cell types or to learn about landmarks from the surrounding cells.

We thank the reviewer for this comment. We have now elaborated in the Discussion at page 13 on the importance of combining ClumpSeq with scRNAseq to reconstruct rare cell type zonation profiles and also contrasted it with other spatial transcriptomics approaches:

“While ClumpSeq directly provides the zonation profiles of cell-type specific genes, its real power of discovery emerges when using the two-step approach of first identifying a set of cell-type specific landmark genes to be used for reconstructing a genome-wide spatial single-cell atlas of the rare cell type (Fig. 1). This enables identifying zoned patterns of genes that are expressed in more than one cell type. An example is Fabp1, a gene that is highly abundant in enterocytes and peaks at the mid-villus zone, but is also zoned in tuft cells towards the villus tip. The expression of Fabp1 in clumps is dominated by the zonation of enterocytes, yet the tuft cell zonation emerges when reconstructing single sequenced tuft cells with the ClumpSeq-identified tuft cell landmark genes (Fig. 4b-d).

“Recently, two powerful methods were developed for measuring spatial information with single-cell resolution, Slide-Seq⁴² and High-definition spatial transcriptomics (HDST⁴³). Similarly to ClumpSeq, both methods need to be integrated with scRNAseq datasets in order to properly analyse the rare cells of interest. However, pre-requirements differ, as Slide-seq and HDST both work on fresh-frozen slides and require barcoding beads on a glass surface, while ClumpSeq uses freshly dissociated tissue with the clumps sorted in 384-well capture plates. Unlike Clumpseq, the spatial transcriptomics methods sequence thin tissue sections that often do not include complete cells, potentially posing challenges in the characterization of individual rare cell types.”

2. I would expect to have the landmarks within the rare cell types. Less differentiated progenitors should be within the crypts, having a higher proportion of transcripts for ribosomal and cell cycle promoting genes, whereas more differentiated cells should express more specialized markers.

We agree with the reviewer that ribosomal and cell cycle genes are more abundant in the crypt vs. villi and can therefore be utilized as landmarks for differentiating single cells between crypts and villi. However, variation of ribosomal genes along the villus is relatively small for secretory cells, limiting their information capacity to localize cells along the villus axis (Figure R1). In contrast, our ClumpSeq analysis enabled extraction of landmark genes with sufficient information to localize the rare secretory cell types, based on the enterocyte transcriptomes.

Figure R1 Heatmap of reconstructed zonation profiles of ribosomal genes for enterocytes, goblet, tuft and enteroendocrine cells. Profiles are normalized to their maximal value across the crypt-villus zones. While levels of ribosomal genes are universally higher in crypt cells, variation along the villus is relatively low for secretory cells.

3. Furthermore, either RNA in situ hybridization or antibody staining are still required to confirm the spatial localisation of the rare cells.

We thank the reviewer for this comment. We have now expanded our study with smFISH validation for 15 additional genes predicted to be zoned according to our ClumpSeq results. These include 6 goblet genes, 4 tuft cell genes and 5 enteroendocrine genes. For goblet cells, we validated the crypt/villus-bottom zoned genes *Spink4*, *Agr2* and *Sox9* and the villus tip-zoned genes *Neat1*, *Cdkn1a* and *Cdh17* (Supplementary Figure 8). For tuft cells, we validated the crypt/villus- bottom zoned *Tuft1* genes *Cirbp* and *Nrep* as well as the villus tip tuft genes *Plek* and *il17rb* (Supplementary Figure 11). For enteroendocrine cells we further validated the expression of *Afp* and *Pyy* in the crypt/villus- bottom and of *Cck*, *Gpx3* and *Nts* in the higher villus zones (Supplementary Figure 15). With the exception of *Cdkn1a* and *Gpx3*, all genes followed the zonation profiles predicted by our Clumpseq reconstruction.

Supplementary Figure 8 Validation of the reconstructed zonation profiles using smFISH. Blue line - smFISH mean expression level, red line - reconstructed profile based on the single cell analysis. Light areas denote the SEM. SmFISH results based on 2 mice, 5 villi per mouse.

Supplementary Figure 11 Quantification of zoned tuft genes smFISH experiment. P value was calculated by Mann Whitney U test. n=3mice.

Supplementary Figure 15 (a) SmFISH quantification of zoned enteroendocrine genes. P value was calculated by Mann Whitney U test. n=3-4 mice. (b,d) Quantification of Pyy⁺ and Afp⁺ enteroendocrine cells in crypt and villus bottom, middle and tip over 3-4 mice. P values are Fisher exact tests for the frequencies of the cells between the two lower zones and two upper zones. Representative smFISH images of the enteroendocrine crypt-zoned gene (c) Pyy (red) and (e) Afp (red).

4. In the abstract, the authors claim: "We uncover immune-modulatory programs in villus tip goblet and tuft cells and heterogeneous migration patterns of enteroendocrine cells." Describing the differential expression of some genes implicated in immune response or targets of interferon signalling does not mean discovering immune-modulatory programs.

We have now changed the sentence:

"We identify elevated expression of immune-modulatory genes in villus tip goblet and tuft cells."

5. Moreover, heterogeneous migration patterns of enteroendocrine cells were already discussed by Gehart and colleagues, 2019.

The landmark paper by Gehart et al. indeed identified both temporal programs of EE cells as well as separation between crypt and villus. Our Clumpseq method complemented this work with a higher spatial resolution along the villus (Figure R2). This enabled identifying spatial trends along the villus, as we highlight for the Sst+ delta cells (Figure 5), as well as for Pyy and Afp, two genes for which we have now added smFISH validations (Supplementary Figure 15b-e).

Figure R2 ClumpSeq identifies spatial zonation trends along the villus axis. The figure shows the results for Sst Spatial distribution of D cells in (a) Gehart et al., Cell 2019. (b) ClumpSeq prediction. (c) Quantification of Sst+ enteroendocrine D-cells in crypt and villus bottom, middle and tip over 3 mice. Fisher exact test for the frequencies of D-cells between the two lower zones and two upper zones $p = 3.8*10^{-5}$ (as reported in Fig 5c in the manuscript). Black crosses in (a) denote the mean over single crypt and villus single cells in Gehart et al. While Sst levels are indeed higher in the villus, the frequency of Sst+ cells significantly declines from the bottom of the villus towards the tip.

We now better clarify and discuss this point on page 12:

"Recently, Gehart et al. uncovered distinct kinetics for different subtypes of enteroendocrine cells as well as different frequencies between crypts and villi³¹. Using ClumpSeq, we were able to provide more detailed spatial information along the villus axis. For example, while Gehart et al. demonstrated that D cells have a late time-stamp and that Sst expression is higher in the villi compared to the crypts, we showed that along the villus, D cells are enriched at the villus bottom and that D cell frequency and Sst expression decreases towards the villus tip."

Other points:

6. The authors should check for the list of “ligands” in Suppl. Fig. 4. Hspa1a (encoding for heat shock protein) and Rps19 (encodes for ribosomal protein) are not ligands.

We thank the reviewer for noting this mistake. We used the list generated by Ramilowsky et al, Nat Comm, 2015¹. We now checked and re-defined the list of ligands and receptors used for the analysis based on the stringent classification of Ramilowsky et. al. This resulted in removal of Hspa1a since it was flagged as ‘not supported by literature’. We did, however leave in Rps19 as it was classified in Ramilowsky et al. as a ligand supported by published papers (PMID: 11733378, 11107061). This filtering reduced the numbers of ligands from 708 to 697 and the numbers of receptors from 691 to 688. We have now updated Supplementary Figure 9 accordingly.

7. In Beumer et al., PYY+ cells were found mostly in the villi. Here, the cells expressing Pyy at the highest level are in the crypts (Suppl. Fig 6). How do the authors explain such discrepancy?

Indeed, ClumpSeq predicted Pyy as a gene that is higher in the crypt/villus bottom zones. We have now validated this result using smFISH (Figure R3 and Supplementary Figure 15b,c shown above in the response to point 3). For this validation, we counted the proportion of Pyy+ cells over 116 crypts and 65 entire villi over 4 mice. Each villus was divided into bottom, middle and tip as follows: Ada⁺ area is defined as ‘Villus tip’, the remaining villus length is divided equally in two. As demonstrated in Figure R3, Pyy is part of that group of genes with discordancy between space and time, with a higher Pyy expression in the crypt and villus bottom zones. This is confirmed by smFISH validation, where the highest % of Pyy+ cells are found to be located indeed at the villus bottom, followed by the crypt, and with lower frequencies in the villus middle and tip zones.

Figure R3 Pyy expression analysis (a) reconstructed temporal (red) and spatial (blue) profiles for Pyy gene (b) Quantification of Pyy+ cells in crypt and villus bottom, middle and tip over 4 mice (116 crypts and 65 villi). Fisher exact test for the frequencies of Pyy+ cells between the two lower zones and two upper zones $p = 1.3 \times 10^{-4}$. (c) Representative smFISH image of Pyy+ enteroendocrine cells. Scale bar 50 μm. Boxes highlight Pyy+ cells, which are concentrated the lower villus zones.

Reviewer #2 (Remarks to the Author):

The present study by Manco et al. introduces ClumpSeq a method for reconstructing spatial zonation of rare cell types in the intestinal epithelium. ClumpSeq involves sequencing of partially digested tissue clumps comprising ~2-10 cells each. The authors had previously inferred spatial zonation of enterocytes representing the most abundant intestinal epithelial cell type (Moor et al., 2018, Cell) by combining single-cell RNA-seq and in situ quantification of landmark genes using smFISH. The enterocyte zonation profiles from this study were now used as a reference, in order to assign each sequenced clump to a spatial zone. By utilizing secretory cell type-specific genes, zonation coordinates of clumps containing a given secretory cell types were used to match single-cell RNA-seq data of the same secretory cell type, permitting the inference of genome-wide spatial zonation profiles for this cell type. ClumpSeq was applied to investigate patterns of transcriptome zonation and enrichment of functional pathways in distinct zones along the crypt-villus axis.

This is very nice study presenting a robust approach for resolving cell type architecture in spatially organized systems.

The experimental data are of very good quality, and the computational analysis pipeline is solid, although it involves some ad hoc parameter choices, which seem somewhat arbitrary.

The core idea of ClumpSeq is not entirely novel. The authors applied a similar strategy in the past to derive zonation of endothelial cells in the liver (Halpern et al., 2018, Nature Biotechnology), where they sequenced pairs of hepatocytes and endothelial cells to derive zonation of the latter based on hepatocyte zoned landmark genes inferred from smFISH (Halpern et al., 2017, Nature). The major novel aspect of the current approach is the computational deconvolution strategy to infer zonation of rare cell types from ClumpSeq, and this approach could be very useful for revealing localization of rare cell types in other spatially organized tissues. With this method the authors report some interesting and novel findings on spatial zonation of goblet, tuft, and enteroendocrine cells.

The manuscript is concise and well written and refers to the relevant single-cell studies on intestinal epithelial cell types.

I have only a few concerns and suggestions for additional analyses to strengthen the paper:

1. The algorithm involves a number of assumptions and ad hoc parameter choices, e.g., for the definition of cell type specific genes or landmark gene. The authors should include a computational analysis to demonstrate the robustness of the inferred zonation coordinates with regard to the choice of these parameters. Another example is the factor used to scale the abundance of secretory cell types in order to select the fraction of clumps along the rays in the PCA to be included into the analysis. This factor was chosen as 2 for pairs, and 3 for large clumps. The latter choice seems quite coarse as these clumps could comprise any number of cells up to ~10, and thus the expected probability

of finding a secretory cell in a clump is expected to vary accordingly. Hence, these quantities should also be subject to a robustness analysis.

We thank the reviewer for this important comment. We have now performed an extensive robustness analysis, varying all of the major parameters used for the reconstruction and assaying the divergence in the resulted reconstructed zonation tables over a range of 0.5-1.5 fold from the parameter value used. We found that our reconstruction remains robust for this wide range of parameters. These new results are described on page 29:

Robustness analysis

In order to assess the robustness of the zonation tables to parameter choices, we reconstructed the single cell zonation tables with modified parameter values and examined the change in the correlation of the zonation profiles centers of masses (COM) between the original and perturbed zonation tables as a function of the size of introduced parameter perturbation. To this end, we performed the following for all parameters pertaining to clump secretory type assignment and subsequent single cell reconstructions: we selected ~75 linearly spaced values for each parameter within the 0.5-1.5 fold range. For some parameters, only integer values could be used. For other parameters, not the entire 0.5-1.5 fold range yielded reconstructions due to absence of marker or landmark genes fitting the criteria for these values. For each parameter, we then generated single cell zonation tables for all the 75 different parameter values. In order to compare these reconstructions to the original zonation table, we calculated the Spearman correlation coefficient of the COMs for the expressed genes (above 5×10^{-6}), between each reconstruction and the original. We denote this quantity for parameter i and perturbed value j - P_{ij} . We then plotted for each parameter i , the values P_{ij} for $j=1:75$. The plotted curves (Supplementary Fig. 4a,5a,6a) were smoothed by applying a moving median filter with window size 10. Due to smoothing, the point (0,1) which indicates that for 0 distance between the perturbed and original parameter, the Spearman correlation P_{ij} is 1, was sometimes shifted. We therefore reintroduced this point after applying the moving median filter.

In order to visualize the differences in the zonation profiles of individual genes between the original and perturbed reconstruction when P_{ij} declines, we selected for each cell type a representative example of a parameter choice that yielded low correlation (P_{ij}) and plotted the original and the perturbed reconstructed zonation profiles for 10 selected genes: 5 crypt genes and 5 tip genes (Supplementary Fig. 4b,5b,6b). The genes shown for these examples were selected as follows: several gene types were excluded (exclusion criteria below), the remaining genes were sorted by COM and top 5 (tip) and bottom 5 (crypt) were auto-selected.

Genes were excluded based on the following criteria, considering the original reconstruction:

1. landmark genes used to generate the original reconstructions
2. low confidence genes - q value above 0.05
3. not highly zoned genes - less than 2 fold dynamic range
4. lowly expressed genes - with maximal normalized expression below 5×10^{-5}

And in Supplementary Figure 4 (goblet cell robustness analysis), 5 (tuft cell robustness analysis) and 6 (enteroendocrine cell robustness analysis):

Supplementary Figure 4 – (a) Correlations between zonation reconstructions as function of parameter perturbation for goblet single cell reconstructions. X axis indicated the distance between the original and perturbed parameter calculated as the difference between the two divided by the original value in percent. The Y axis indicates the Spearman correlation coefficient between the centers of masses (COMs) of the original reconstruction and perturbed reconstruction (See methods for details). Parameter are detailed in Supplementary table 11. Red circle marked with red arrow indicates an example of a perturbed reconstruction for which correlation with the original reconstruction was lower. In panels b we show examples for individual zonation profiles in the original (blue lines) compared to the indicated perturbed reconstruction (dashed red lines). For this purpose, 5 tip and 5 crypt zoned, highly expressed genes not used as landmarks were selected (See methods).

Supplementary Figure 5 – (a) Correlations between zonation reconstructions as function of parameter perturbation for tuft single cell reconstructions. X axis indicated the distance between the original and perturbed parameter calculated as the difference between the two divided by the original value in percent. The Y axis indicates the Spearman correlation coefficient between the centers of masses (COMs) of the original reconstruction and perturbed reconstruction (See methods for details). Parameter are detailed in Supplementary table 11. Red circle marked with red arrow indicates an example of a perturbed reconstruction for which correlation with the original reconstruction was lower. In panels b we show examples for individual zonation profiles in the original (blue lines) compared to the indicated perturbed reconstruction (dashed red lines). For this purpose, 5 tip and 5 crypt zoned, highly expressed genes not used as landmarks were selected (See methods).

Supplementary Figure 6 – (a) Correlations between zonation reconstructions as function of parameter perturbation for enteroendocrine single cell reconstructions. X axis indicated the distance between the original and perturbed parameter calculated as the difference between the two divided by the original value in percent. The Y axis indicates the Spearman correlation coefficient between the centers of masses (COMs) of the original reconstruction and perturbed reconstruction (See methods for details). Parameter are detailed in Supplementary table 11. Red circle marked with red arrow indicates an example of a perturbed reconstruction for which correlation with the original reconstruction was lower. In panels b we show examples for individual zonation profiles in the original (blue lines) compared to the indicated perturbed reconstruction (dashed red lines). For this purpose, 5 tip and 5 crypt zoned, highly expressed genes not used as landmarks were selected (See methods).

We have also added a new supplementary table (Supplementary Table 11) detailing the parameters assayed in Supplementary Figure 4, 5 and 6.

2. Since the authors are able to record index data with their sequencing approach, this factor could be modeled more systematically given the number of cells in each clump estimated from DNA staining. My worry is that a too conservative cutoff potentially eliminates clumps with less pronounced expression of cell type specific genes, which could be localized to specific zones, e.g., progenitor stages, naively expected to be localized preferentially to the crypt bottom.

This is a great idea, but unfortunately, we did not record index data for sequenced clumps. As for the reviewers concern regarding extensive loss of crypt clumps due to lower expression of type markers, we avoid this by selecting crypt and villus secretory containing clumps separately. We first assign each clump with a zone and then select the secretory cell-containing crypt clumps as the top type marker expressing clumps out of only the crypt clumps. The proportion selected is derived from our measured abundance of each cell type in the crypts. Similarly, we select secretory containing villus clumps out of the group of all villus clumps. We describe this procedure in further detail in the methods section on page 21:

“To minimize miss-classification we therefore sorted the clumps on each ray in descending order according to the distance from the origin, and included a fraction that matches the abundance of this cell type in the tissue. To estimate these abundances, we measured the proportions of each secretory cell type in both crypts and villi of the jejunum out of all cells (Supplementary Table 8). Measurements were performed by imaging the tuft cell marker *Dclk1*, the enteroendocrine cell marker *Chga* and the goblet cell marker *Cla1*. For Paneth cells, data was taken from Elmes M.J.⁵². For final thresholds, measured proportions for crypts and villi were multiplied by 2 for pairs and by 3 for larger clumps, to represent the higher probability to capture rare events in clumps.”

2. The authors validate the spatial reconstruction of goblet cells by smFISH for a number of markers with different zonation patterns, which very nicely shows the quality of the ClumpSeq inference. However, for enteroendocrine and tuft cells they only include smFISH validation for one or two genes, respectively. Please include a few more genes with complementary zonation patterns for each cell type.

We have now expanded our study with smFISH validation for 15 additional genes predicted to be zoned according to our ClumpSeq results. These include 6 goblet genes, 4 tuft cell genes and 5 enteroendocrine genes. For goblet cells, we validated the crypt/bottom villus zoned genes *Spink4*, *Agr2* and *Sox9* and the villus tip-zoned genes *Neat1*, *Cdkn1a* and *Cdh17* (Supplementary Figure 8). For tuft cells, we validated the crypt/bottom villus zoned Tuft1 genes *Cirbp* and *Nrep* as well as the villus tip tuft genes *Plek* and *il17rb* (Supplementary Figure 11). For enteroendocrine cells we further validated the expression of *Afp* and *Pyy* in the crypt/bottom villus

and of *Cck*, *Gpx3* and *Nts* in the higher villus zones (Supplementary Figure 15). With the exception of *Cdkn1a* and *Gpx3*, all genes followed the zonation profiles predicted by our Clumpseq reconstruction.

Supplementary Figure 8 Validation of the reconstructed zonation profiles using smFISH. Blue line - smFISH mean expression level, red line - reconstructed profile based on the single cell analysis. Light areas denote the SEM. SmFISH results based on 2 mice, 5 villi per mouse.

Supplementary Figure 11 Quantification of zoned tuft genes smFISH experiment. P value was calculated by Mann Whitney U test. n=3mice.

Supplementary Figure 15 (a) SmFISH quantification of zoned enteroendocrine genes. P value was calculated by Mann Whitney U test. n=3-4 mice. (b,d) Quantification of Pyy⁺ and Afp⁺ enteroendocrine cells in crypt and villus bottom, middle and tip over 3-4 mice. P values are Fisher exact tests for the frequencies of the cells between the two lower zones and two upper zones. Representative smFISH images of the enteroendocrine crypt-zoned gene (c) Pyy (red) and (e) Afp (red).

3. Is it possible to reconstruct a differentiation program of goblet cells from their zonation profile? Naively, I would expect that maturation concurs with migration upward along the crypt-villus axis, and thus the differentiation stage should be a correlate of the zonation coordinate. This should be tested for the other secretory cell types as well, beyond the timestamp analysis presented for enteroendocrine cells.

We thank the reviewer for this interesting point. We believe that the question of whether zoned secretory cell types are distinctly different lineages, or rather represent trans-differentiating cells is important and we have now performed analyses along several directions. As we show below, the results for Tuft cells are unclear and should be resolved in future work by lineage tracing models. We therefore decided not to include these results in our paper and discussed these future directions in the text.

For studying differentiation trajectories of goblet and tuft populations, we performed computational and smFISH analysis (Figures R4, R5). In particular, we used Monocle3 for pseudotime analysis, examined the co-expression of crypt/bottom villus-zoned genes and villus tip-zoned genes and validated the results with smFISH experiments. For goblet cells, Monocle3 showed a progressive differentiation as the cells migrate along the villus axis (Figure R4a,b). Moreover, the co-expression analysis shows the transition of expression between crypt and villus tip markers as they move upwards (Figure R4c,d). Finally, we also validated this transition using the smFISH for Spink4 and Ido1. Crypt goblet cells uniquely expressed Spink4 (Figure R4e). At the villus bottom, we observed co-expression of Spink4 and Ido1, with a gradual decrease in the levels of Spink4 and increase in the levels of Ido1 with villus coordinate (Figure R4e). Altogether, these results support a gradual but continuous differentiation of goblet cells from crypt to villus tip.

Figure R4 Goblet trans-differentiation analysis (a-b) Monocle trajectories of the single sequenced goblet cells colored by Monocle pseudotime (a) and by our reconstructed crypt-villus zone (b). (c-d) Scatter plots of the normalized expression of crypt (*Agr2* and *Spink4*) vs tip (*Ido1*) goblet specific genes. Goblet cells at the mid-villus zones show co-expression of crypt and tip markers, indicative of gradual transition between the two cell states. (e) Representative smFISH image showing the co-expression of crypt and tip marker genes at the villus bottom. In the whole image, *Clca1* (magenta), E-cadherin antibody (white). The insert shows *Spink4* mRNAs (red dots) and *Ido1* mRNAs (Cyan dots). Scale bar 20 μm . Red arrow marks a *Spink4*+*Ido1*- gene in the crypt, cyan arrows mark cells co-expressing both *Ido1* and *Spink4*.

The question of continuous trans-differentiation is more ambiguous for Tuft cells. Our Monocle3 analysis yielded a bifurcated trajectory between tuft1 and tuft2 states (Figure R5a,b). Supporting this result, only ~30% of crypt tuft cells expressed the tuft1 gene *Nrep* (Figure R5g). In contrast, when we correlated tuft1 and tuft2 genes we did not find that they were mutually exclusive (Figure R5c-f). These contrasting results prohibited converging on a clear picture of whether tuft1 and tuft2 are distinct lineages, or whether tuft1 transition into tuft 2 cells.

Figure R5 Tuft trans-differentiation analysis Monocle trajectory analysis colored by Monocle pseudotime (a), and by our reconstructed crypt-villus zones (b). (c-f) Scatter plots of the normalized expression of tuft1 (Nrep and Cirbp) vs tuft2 (Plek and Ptgs1) tuft specific genes. (g) Representative smFISH image showing of Nrep+ cells (red) in the crypt.

For a more robust result, a lineage tracing experiment for tuft cells would be needed. Unfortunately, such mouse models with a tuft1-gene promoter driven Cre is not yet available. We now discuss this future direction on page 13:

“Our zonation results indicated that both goblet cells and tuft cells express immune-modulatory programs at the villus tip. We further showed that the neuronal-like tuft1 program is zoned towards the crypt and villus bottom, whereas the immune-related tuft2 program is zoned towards the tip. It will be important to utilize lineage tracing mouse models, such as in Beumer et al.³¹ to examine whether these zoned cell states represent continuous trans-differentiation or rather distinct lineages that settle in different crypt-villus coordinates.”

4. It would be helpful to include a plot showing the expected frequency of each rare secretory cell type along the crypt-villus axis derived from the ClumpSeq data. This has been somewhat addressed for some of the enteroendocrine cells, but a summary figure showing the frequency of all secretory cell types as a function of the crypt-villus coordinate derived from the enterocyte profiles would be informative.

We now show cell type frequencies by zone in Supplementary figure 3c. This figure was generated using our clumps zonation table, coarse grained into 4 zones. The reason for the coarse-graining is the rarity of tuft and enteroendocrine clumps along the villi.

Supplementary Figure 3c - Frequency of clump types in the crypt and 3 villus zones. The calculation was performed on the clumps zonation table Table S2, coarse grained into 4 zones.

5. In important goal of spatial gene expression analysis is the inference of molecular interactions between different cell types. Is there spatial co-localization of specific zoned sub-types of different populations, and is it possible to infer in silico, if these sub-types are interacting through specific molecular pathways? For example, are there ligand-receptor pairs suggesting the interaction of enteroendocrine cells confined to the lower crypt with crypt-bottom enterocytes? Could these interactions be involved in controlling the local maturation of one or the other cell type?

We thank the reviewer for the excellent suggestion that prompted us to perform ligand-receptor interaction analysis between zoned epithelial and mesenchymal cells.

We added a paragraph on page 11 describing this:

Zone-specific interactions between epithelial and mesenchymal cells

The zoned expression programs ClumpSeq suggested that secretory cells might be interacting with other zoned small intestinal cell types. Both enterocytes⁷ and mesenchymal cells⁴⁰ were shown to exhibit strongly zoned gene expression signatures. We therefore sought to identify potential zone-dependent interactions. We analyzed a database of ligands and receptors²⁰ and identified pairs in which a ligand was enriched in one cell population and the matching receptor was enriched in another (Supplementary Table 10). We further examined ligand-receptor pairs enriched in either the crypt or bottom villus zones (Supplementary Fig 17a) or in the villus tip zone (Supplementary Fig 17b). Our analysis revealed classic interactions such as crypt telocyte Rspo genes and crypt enterocyte Lgr5 and Lgr4, as well as signaling from tuft cells through the zoned ligands Inhbc, Dll3 and Ccl5 as well as signaling by zoned enteroendocrine cells (Supplementary Fig 17a). The tip zone included the autocrine Il25-Il17rb circuit operating in tuft, as well as signaling from K enteroendocrine cells through the zoned Efna1, Ccl28 and Gip (Supplementary Fig 17b).

The analysis results are shown in the new Supplementary Table 10 and in the new Supplementary Figure 17 a and b:

a Interaction villus bottom

b Interaction villus tip

Supplementary Figure 17. Networks of enriched ligand-receptor interactions between epithelial and mesenchymal cells (a) at the villus bottom and (b) at the villus tip. Expression of either the ligand or the receptor above 2×10^{-5} and $Z_{interaction}$ higher than 5 (Methods).

We also added a description in the Methods section:

Ligand-receptor analysis

Ligand-receptor analysis was performed similar to Bahar-Halpern et al.⁴⁰. We performed the analysis between and within zonated epithelial and mesenchymal cell types. We used the zonated secretory cell expression reconstructed with Clumpseq and previously published datasets for zonated enterocytes and mesenchymal cells^{7,40}. A list of ligand-receptor pairs was extracted from Ramilowski et al²⁰ (697 unique ligands and 688 unique receptors). For each gene g and each cluster c we calculated the average expression x_g^c . We then computed a Z-score, Z_g^c , representing the enrichment of each ligand and receptor in each cell type:

$$Z_g^c = \frac{x_g^c - \text{mean}(x_g^c)}{\text{std}(x_g^c)}$$

Where the mean and standard deviations were compute over all cell types. We next defined an interaction score as: $Z_{interaction} = \sqrt{(Z_L^{C1})^2 + (Z_R^{C2})^2}$

Where Z_L^{C1} is the ligand Zscore for cell type C_1 , and Z_R^{C2} is the receptor Zscore for cell type C_2 . The resulting list of interactions was filtered per fraction of cells expressing either the ligand or the receptor (>0.05) and the $Z_{interaction}$ was above 2.

Cytoscape⁵³ was used to visualize bottom and tip interactions. We selected only ligands and receptors with an average expression of either the ligand or the receptor above $2 * 10^{-5}$ and $Z_{interaction}$ higher than 5. For interactions that occur at the crypt or villus bottom zones (Supplementary Fig 17a), we considered EC, D, L, X and N enteroendocrine cells and crypt-villus bottom goblet, tuft, enterocytes and telocytes. For interactions at the villus tip (Supplementary Fig 17b), EC, I, L and N enteroendocrine cells and villus tip goblet, tuft, enterocytes and telocytes.

6. The authors argue based on the observed anticorrelation of crypt and villus tip enterocyte markers, as well as based on the expression of crypt enterocyte markers and Paneth cell markers, that clumps are a result of incomplete digestion rather than forming from individual cells in solution. However, although these data indicate that the major contribution comes from actual tissue clumps, the "background" contribution from artificial clumps emerging after complete dissociation from single cells should be analyzed. This should be done experimentally by quantifying the number of clumps reshaping in solution after longer digestion using FACS.

We thank the reviewer for the excellent suggestion. Indeed, the number of clumps could potentially increase during the sorting time as a result of unspecific binding between single cells. To exclude that this is a major factor in our experiments, we analyzed the % of clumps in solution immediately after dissociation and after letting the samples on ice for 3h (which represents the longest timing of sorting we used) using both, the single cell and clumps dissociation protocol. We ran the analysis at two different time points (0h and after 3h) using the SORP-FACSARIAII machine.

As can be seen in Supplementary Figure 2, the % of clumps is 2.1% using the single-cell dissociation protocol and 7.79% when using the clumps dissociation protocol. Importantly, however, the % of clumps had little variation after 3h on ice (7.79% of clumps at time 0h vs 7.95% of clumps after 3h, $p = 0.12$).

These results are described on page 4:

To further verify that single cells do not form clumps in solution, we demonstrated that the percentages of clumps immediately after tissue dissociation and after 3 hours of incubation had little variation (clumps dissociation protocol, 7.79% of clumps at time 0h vs 7.95% of clumps after 3h, $p = 0.12$, Supplementary Fig. 2).

and in Supplementary Figure 2:

Supplementary Figure 2 Background analysis for dissociation protocol (a-b) Representative histograms of the Hoechst DNA content staining at time 0h (red curve) and 3 h (green curve) for the single-cell dissociation protocol (a) and clump dissociation protocol (b). (c) Quantification of clumps immediately after dissociation for single-cell and clumps protocol at time 0h and after 3h, $n=3$ mice. (sc protocol p value by t test = 0.03; clumps protocol p value by t test = 0.12).

Reviewer #3 (Remarks to the Author):

In the submitted manuscript Rita Manco, Inna Averbukh and colleagues describe a new approach – ClumpSeq - to catalog rare cell in the small intestine. ClumpSeq is an interesting technique, which together with additional approaches can be used to infer the spatial situation of different cell types in the tissue. Using this method, the authors succeeded to capture rare cells as goblet cells, enteroendocrine and tuft cell and distinctively infer their gene expression profiles specific to the different zonation. The study is well performed, claims are supported by data. The manuscript may represent an important method/resource paper for scientists interesting in spatial and possibly also functional characterization of rare cell types at single cell level. Nevertheless, there are some issues that should be addressed in a revision. Majority of them do not point to weak or critical points but represent questions that arose during the reading of the manuscript as result of the curiosity. Hopefully addressing them may be helpful also for the authors.

1. Combination of ClumpSeq with LCM and conventional scRNA-seq provides a more detailed spatially corrected transcriptomic profile that could help predict/hypothesize more complex cellular features. An intriguing concept that could be addressed in this work is to try and provide more explanation regarding cellular dynamics. For instance, can the authors analyze whether villus-tip differentiated cells (goblet, tuft) are derived from the same cell types previously located at the villus bottom/crypt or there are different progenitors that can give rise to different bottom and top cells? Do the tip cells differentiate in the tip or they can be dynamic while migrating towards the top, acquiring different identities? Can some cellular trajectories be charted for rare epithelial cells describing the development of distinct spatial sub-types of rare cells?

We thank the reviewer for this excellent point. We believe that the question of whether zonated secretory cell types are distinctly different lineages, or rather represent trans-differentiating cells is important and we have now performed analyses along several directions. As we show below, the results for Tuft cells are unclear and should be resolved in future work by lineage tracing models. We therefore decided not to include these results in our paper and discussed these future directions in the text.

For studying differentiation trajectories of goblet and tuft populations, we performed computational and smFISH analysis (Figures R4, R5). In particular, we used Monocle3 for pseudotime analysis, examined the co-expression of crypt/bottom villus-zonated genes and villus tip-zonated genes and validated the results with smFISH experiments. For goblet cells, Monocle3 showed a progressive differentiation as the cells migrate along the villus axis (Figure R4a,b). Moreover, the co-expression analysis shows the transition of expression between crypt and villus tip markers as they move upwards (Figure R4c,d). Finally, we also validated this transition using the smFISH for Spink4 and Ido1. Crypt goblet cells uniquely expressed Spink4 (Figure R4e). At the villus bottom, we observed co-expression of Spink4 and Ido1, with a gradual decrease in the levels of Spink4 and increase in the

levels of *Ido1* with villus coordinate (Figure R4e). Altogether, these results support a gradual but continuous differentiation of goblet cells from crypt to villus tip.

Figure R4 Goblet trans-differentiation analysis (a-b) Monocle trajectories of the single sequenced goblet cells colored by Monocle pseudotime (a) and by our reconstructed crypt-villus zone (b). (c-d) Scatter plots of the normalized expression of crypt (*Agr2* and *Spink4*) vs tip (*Ido1*) goblet specific genes. Goblet cells at the mid-villus zones show co-expression of crypt and tip markers, indicative of gradual transition between the two cell states. (e) Representative smFISH image showing the co-expression of crypt and tip marker genes at the villus bottom. In the whole image, *Clca1* (magenta), E-cadherin antibody (white). The insert shows *Spink4* mRNAs (red dots) and *Ido1* mRNAs (Cyan dots). Scale bar 20 μm . Red arrow marks a *Spink4*+*Ido1*- gene in the crypt, cyan arrows mark cells co-expressing both *Ido1* and *Spink4*.

The question of continuous trans-differentiation is more ambiguous for Tuft cells. Our Monocle3 analysis yielded a bifurcated trajectory between tuft1 and tuft2 states (Figure R5a,b). Supporting this result, only ~30% of crypt tuft cells expressed the tuft1 gene *Nrep* (Figure R5g). In contrast, when we correlated tuft1 and tuft2 genes we did not find that they were mutually exclusive (Figure R5c-f). These contrasting results prohibited converging on a clear picture of whether tuft1 and tuft2 are distinct lineages, or whether tuft1 transition into tuft 2 cells.

Figure R5 Tuft trans-differentiation analysis Monocle trajectory analysis colored by Monocle pseudotime (a), and by our reconstructed crypt-villus zones (b). (c-f) Scatter plots of the normalized expression of tuft1 (Nrep and Cirbp) vs tuft2 (Plek and Ptg1) tuft specific genes. (g) Representative smFISH image showing of Nrep+ cells (red) in the crypt.

For a more robust result, a lineage tracing experiment for tuft cells would be needed. Unfortunately, such mouse models with a tuft1-gene promoter driven Cre is not yet available. We now discuss this future direction on page 13:

“Our zonation results indicated that both goblet cells and tuft cells express immunomodulatory programs at the villus tip. We further showed that the neuronal-like tuft1 program is zoned towards the crypt and villus bottom, whereas the immune-related tuft2 program is zoned towards the tip. It will be important to utilize lineage tracing mouse models, such as in Beumer et al.³¹ to examine whether these zoned cell states represent continuous trans-differentiation or rather distinct lineages that settle in different crypt-villus coordinates.”

2. Can be data acquired in this manuscript somehow integrated with previous datasets determining intestinal mesenchymal cells (McCarthy et al., 2020) or zoned intestinal mesenchyme generated previously by Itzkovitz lab (Bahar Halpern et al. 2020). Such an integratory approach may reveal a specific, rare/secretory cells-mesenchymal

interactions. For example, SuppFig.4b indicates high expression of *Pdgfra* by villus-tip goblet cell. At the villus *Pdgfra* mesenchymal cells are located (McCarthy et al., 2020). Is it just a co-incidence? Or based on Figure 3e, it seems that *Ido1* is also expressed in mesenchymal cells that co-express *Cla1*. Can authors comment this?

We thank the reviewer for the excellent suggestion that prompted us to perform ligand-receptor interaction analysis between zonated epithelial and mesenchymal cells.

We added a paragraph on page 11 describing this:

Zone-specific interactions between epithelial and mesenchymal cells

The zonated expression programs ClumpSeq suggested that secretory cells might be interacting with other zonated small intestinal cell types. Both enterocytes⁷ and mesenchymal cells⁴⁰ were shown to exhibit strongly zonated gene expression signatures. We therefore sought to identify potential zone-dependent interactions. We analyzed a database of ligands and receptors²⁰ and identified pairs in which a ligand was enriched in one cell population and the matching receptor was enriched in another (Supplementary Table 10). We further examined ligand-receptor pairs enriched in either the crypt or bottom villus zones (Supplementary Fig 17a) or in the villus tip zone (Supplementary Fig 17b). Our analysis revealed classic interactions such as crypt telocyte *Rspo* genes and crypt enterocyte *Lgr5* and *Lgr4*, as well as signaling from tuft cells through the zonated ligands *Inhbc*, *Dll3* and *Ccl5* as well as signaling by zonated enteroendocrine cells (Supplementary Fig 17a). The tip zone included the autocrine *Il25-Il17rb* circuit operating in tuft, as well as signaling from K enteroendocrine cells through the zonated *Efna1*, *Ccl28* and *Gip* (Supplementary Fig 17b).

The analysis results are shown in the new Supplementary Table 10 and in the new Supplementary Figure 17 a and b:

a Interaction villus bottom

b Interaction villus tip

Supplementary Figure 17. Networks of enriched ligand-receptor interactions between epithelial and mesenchymal cells (a) at the villus bottom and (b) at the villus tip. Expression of either the ligand or the receptor above 2×10^{-5} and $Z_{interaction}$ higher than 5 (Methods).

We also added a description in the Methods section:

Ligand-receptor analysis

Ligand-receptor analysis was performed similar to Bahar-Halpern et al.⁴⁰. We performed the analysis between and within zonated epithelial and mesenchymal cell types. We used the zonated secretory cell expression reconstructed with Clumpseq and previously published datasets for zonated enterocytes and mesenchymal cells^{7,40}. A list of ligand-receptor pairs was extracted from Ramilowski et al²⁰ (697 unique ligands and 688 unique receptors). For each gene g and each cluster c we calculated the average expression x_g^c . We then computed a Z-score, Z_g^c , representing the enrichment of each ligand and receptor in each cell type:

$$Z_g^c = \frac{x_g^c - \text{mean}(x_g^c)}{\text{std}(x_g^c)}$$

Where the mean and standard deviations were compute over all cell types. We next defined an interaction score as: $Z_{interaction} = \sqrt{(Z_L^{C1})^2 + (Z_R^{C2})^2}$

Where Z_L^{C1} is the ligand Zscore for cell type C_1 , and Z_R^{C2} is the receptor Zscore for cell type C_2 . The resulting list of interactions was filtered per fraction of cells expressing either the ligand or the receptor (>0.05) and the $Z_{interaction}$ was above 2.

Cytoscape⁵³ was used to visualize bottom and tip interactions. We selected only ligands and receptors with an average expression of either the ligand or the receptor above $2 * 10^{-5}$ and $Z_{interaction}$ higher than 5. For interactions that occur at the crypt or villus bottom zones (Supplementary Fig 17a), we considered EC, D, L, X and N enteroendocrine cells and crypt-villus bottom goblet, tuft, enterocytes and telocytes. For interactions at the villus tip (Supplementary Fig 17b), EC, I, L and N enteroendocrine cells and villus tip goblet, tuft, enterocytes and telocytes.

3. What exactly does it mean “immune-specialization of goblet cells at the tip of the villus” except of enhanced IFN-response? Is it such a response specific to goblet cells at the villus-tip? Or other villus tip cells (incl. enterocytes) are also affected? Recent data indicated possible impact of IFN γ on cellular differentiation in the crypt (Biton et al., 2018), including restriction of secretory cell differentiation (Sato et al., 2020). Are any of these mechanisms relevant also for tip-villus goblet cells?

Indeed, both enterocytes and goblet cells express the interferon gamma response at the villus tip (Supplementary Table 4). We have now included an analysis of the expression of the hallmark interferon gamma response gene set (Supplementary Figure 10). Interestingly, each cell type expresses distinct genes from this pathway. For example, *Ido1* is specific for goblet cells, while enterocytes express *Ddx58*, receptor responsible for IFN1 response and essential for recognizing cells that have been infected by a virus. The different clusters suggest that all cells at the tip respond to IFN γ , but through distinct mechanisms.

We now included this analysis in the manuscript on page 6:

Interferon-alpha (IFN α) and interferon-gamma (IFN γ) responses were also enriched in tip enterocytes, however the identity of the tip-enterocyte and tip-goblet cell genes was largely distinct (Supplementary Fig. 10). Tip goblet cell IFN γ genes included the immune checkpoint target gene *Ido1* (Fig. 3a,e), previously shown to have immunosuppressive effects²². Tip enterocyte IFN γ genes included the viral response receptor *Ddx58*²³. The different clusters suggest that all cells at the tip react to IFN γ , but through distinct mechanisms that are cell-type specific.

The results are shown in Supplementary Figure 10:

Supplementary Figure 10 Interferon-gamma response genes are differentially expressed between villus tip enterocytes, and villus tip goblet cells.

4. Can the authors identify progenitors vs. newly differentiated cells in the crypt? For instance, in page 8 the authors state that “Tuft cells at the crypt expressed the transcription factor Sox4”. Are these Tuft cells progenitors or differentiated Tuft cells? Based on Figure 4a, is it correct that they are still proliferative, based on expression of *Ccnd1*? What about other rare cell types?

We have now analyzed *Mki67* expression in tuft cells with smFISH (Figure R6a). In the crypt, very few *Dclk1*⁺ cells expressed *Mki67*, as also seen in our single cell data (Figure R6b). SmFISH for Sox4, *Mki67* and *Dclk1* revealed mutually exclusive expression of Sox4 and *Mki67*. This is in accordance with previously published data demonstrating that *Dclk1*⁺ tuft cells are, indeed, post-mitotic cells^{2,3}.

Figure R6 *Dclk1*⁺ tuft cells in the crypt are post-mitotic cells (a) representative smFISH images of *Dclk1* (white dots) Sox4 (red dots) and *Mki67* (green dots). Crypt Sox4⁺ cells do not co-express *Mki67* and are therefore post-mitotic. Scale bar 10 μ m. (b) tSNE plots showing the *Mki67* expression from the single cell sequencing.

5. Are there any transcriptional differences between the proposed slow vs. fast migrating enteroendocrine cells that could explain their different behavior? Perhaps any gene signature featuring migratory characteristics? In addition, staining for villus top enteroendocrine cells could also be interesting to add. Can authors speculate what could be the functional implication of these differences?

We thank the reviewer for this excellent suggestion. We now looked for differences in gene expression between the slowly-migrating Sst D cells and the more rapidly migrating Tac1⁺ and Gcg⁺ cells. We used GOrilla software (<http://cbl-gorilla.cs.technion.ac.il/>) to analyze the differentially-expressed genes (2.5 fold higher expressed and with a p value < 0.05) between these two groups of cells. This analysis revealed several interesting genes elevated in D cells that are related to binding to the extra-cellular matrix and to neighboring epithelial cells. We now discuss this result in page 11:

Finally, we examined the differences in gene expression between D cells, predicted to migrate slowly, and Tac1⁺ EC cells and Gcg⁺ L cells, predicted to migrate more rapidly³¹. Gene set enrichment analysis³⁵ revealed that D cells were enriched in GO programs related to adhesion and migration (Supplementary Table 9). In particular, they highly expressed Itbg5 and Emp2, genes involved in cell-matrix adhesion^{36,37}, Mylk, important for adhesion disassembly mechanism³⁸, and Csf1 and its receptor Csf1r, implicated in the formation of unstable interactions³⁹ (Supplementary Fig 16). The slower migration of D cells may thus be associated with more stable interactions with the extracellular matrix, or to more unstable interactions with neighboring epithelial cells.

and add the differential gene expression analysis as Supplementary Figure 16:

Supplementary Figure 16 Differential expression analysis between slow D cells and fast EC and L cells. Volcano plot of DGE between D cells and EC and L cells. Labeled dots are selected differentially expressed genes related to GO adhesion program.

We also discussed this in page 12:

Somatostatin inhibits the secretion of other hormones, such as Cck⁴¹. Moreover, using our ligand-receptor analysis we found that most enteroendocrine cells express the somatostatin receptors (Supplementary Fig. 17a). Stalled D cells in the villus bottom might therefore serve to prevent secretion of some of these hormones specifically at the lower villus zones.

We also added new smFISH validations for 3 new enteroendocrine tip genes, as shown in Supplementary Figure 15a

Supplementary Figure 15 (a) SmFISH quantification of zoned enteroendocrine genes. P value was calculated by Mann Whitney U test. n=3-4 mice.

6. It would be helpful to add to Figure 4b/c, a staining with lower magnification to appreciate the expression pattern of Fabp1 along the crypt/villus axis.

We have now added the low magnification as **Supplementary Figure 12**

Supplementary Figure 12 whole images of the representative smFISH in figure 4 of the main manuscript (a) villus bottom (b) villus tip. Scale bar 30 μm . Arrows mark the tuft cells.

7. It might be worth to compare/discuss ClumpSeq (pipeline) with other single cell approaches allowing spatial transcriptomics as for example Slide-seq (Rodrigues et al.,

2019). Maybe some table indicating pre-requirements/resolution/sensitivity etc. would be nice.

We now discuss this on page 13:

“Recently, two powerful methods were developed for measuring spatial information with single-cell resolution, Slide-Seq⁴² and High-definition spatial transcriptomics (HDST⁴³). Similarly to ClumpSeq, both methods need to be integrated with scRNAseq datasets in order to properly analyse the rare cells of interest. However, pre-requirements differ, as Slide-seq and HDST both work on fresh-frozen slides and require barcoding beads on a glass surface, while ClumpSeq uses freshly dissociated tissue with the clumps sorted in 384-well capture plates. Unlike Clumpseq, the spatial transcriptomics methods sequence thin tissue sections that often do not include complete cells, potentially posing challenges in the characterization of individual rare cell types.”

References

1. Ramiłowski, J. A. *et al.* A draft network of ligand-receptor-mediated multicellular signalling in human. *Nat. Commun.* **6**, (2015).
2. Gerbe, F. *et al.* Distinct ATOH1 and Neurog3 requirements define tuft cells as a new secretory cell type in the intestinal epithelium. *J. Cell Biol.* **192**, 767–780 (2011).
3. May, R. *et al.* Doublecortin and CaM kinase-like-1 and leucine-rich-repeat-containing G-protein-coupled receptor mark quiescent and cycling intestinal stem cells, respectively. *Stem Cells* **27**, 2571–2579 (2009).

Reviewers' Comments:

Reviewer #1:

Remarks to the Author:

I have no further comments. All concerns have been addressed.

Reviewer #2:

Remarks to the Author:

All of my previous concerns were thoroughly addressed in the revised manuscript, which is now ready for publication.

Reviewer #3:

Remarks to the Author:

The authors have addressed all my comments more than sufficiently. I believe that ClumpSeq will become an important approach to analyze rare cell types not only in the intestine. My congratulations to authors for the nice work.